# Hydrophobicity of arginine leads to reentrant liquid-liquid phase separation behaviors of arginine-rich proteins

Yuri Hong [1,2], Saeed Najafi[3], Thomas Casey[3], Joan-Emma Shea[3] ✉, Song-I Han [3,4] ✉ & Dong Soo Hwang [1,2] ✉

Intrinsically disordered proteins rich in cationic amino acid groups can undergo Liquid-Liquid Phase Separation (LLPS) in the presence of charge-balancing anionic counterparts. Arginine and Lysine are the two most pre-valent cationic amino acids in proteins that undergo LLPS, with arginine-rich proteins observed to undergo LLPS more readily than lysine-rich proteins, a feature commonly attributed to arginine's ability to form stronger cation-π interactions with aromatic groups. Here, we show that arginine's ability to promote LLPS is independent of the presence of aromatic partners, and that arginine-rich peptides, but not lysine-rich peptides, display re-entrant phase behavior at high salt concentrations. We further demonstrate that the hydrophobicity of arginine is the determining factor giving rise to the reen-trant phase behavior and tunable viscoelastic properties of the dense LLPS phase. Controlling arginine-induced reentrant LLPS behavior using tempera-ture and salt concentration opens avenues for the bioengineering of stress-triggered biological phenomena and drug delivery systems.

Liquid−liquid phase separation (LLPS) is an important organizing principle for biomolecular condensates that compartmentalize spe-cific metabolites without a biological membrane[1,2]. This phenomenon involves the phase separation of macromolecules into two immiscible liquid phases − a dense phase rich in macromolecules (the coacervate phase) and a dilute phase in which the macromolecules are depleted (the supernatant phase) (Fig. 1a). The simplest form of LLPS involves simple or complex coacervation driven by electrostatic interactions. This LLPS is readily realized in polyelectrolyte and polyampholyte systems[3,4]. Naturally occurring proteins however have sequences that are more complex, and LLPS is driven not only by electrostatic cation/anion interactions, but by a host of additional attractive interactions, including cation−π interactions, hydrophobic interactions, π−π inter-actions, hydrogen bonding, and van der Waals interactions[5,6]. Both globular and intrinsically disordered proteins (IDPs) have the ability to

form LLPS, but IDPs account for a large proportion of biological LLPS[7]. IDPs are commonly enriched in charged, aromatic, and polar amino acids, while their structural flexibility makes IDPs more likely to form multivalent interactions[8,9]. The amino acid composition of IDPs tunes the physical property of the dense coacervate phase, such as diffu-sivity, interfacial tension, density, viscoelasticity, and exchange dynamics in the resulting dense coacervate phase[10,11]. These properties of LLPS are key to the bioengineering application of LLPS, including as delivery agents for protein-based drugs and mRNA vaccines[12–14]. Thus, understanding the role of amino acid composition in tuning the per-tinent physical properties of the coacervate phase is important.

Among the different amino acids, arginine is considered an important driver for LLPS[10,15–19]. Considering that the release of mRNA-based cargo is an important application outlook of LLPS, it is note-worthy that arginine is abundant in RNA-binding proteins (RBPs) that

[1]School of Interdisciplinary Bioscience and Bioengineering, Pohang University of Science and Technology (POSTECH), Pohang 37673, Republic of Korea. [2]Division of Environmental Science and Engineering, Pohang University of Science and Technology (POSTECH), Pohang 37673, Republic of Korea. [3]Department of Chemistry & Biochemistry, University of California, Santa Barbara, CA 93106, USA. [4]Department of Chemical Engineering, University of California, Santa Barbara, CA 93106, USA. ✉e-mail: shea@chem.ucsb.edu; songi@chem.ucsb.edu; dshwang@postech.ac.kr

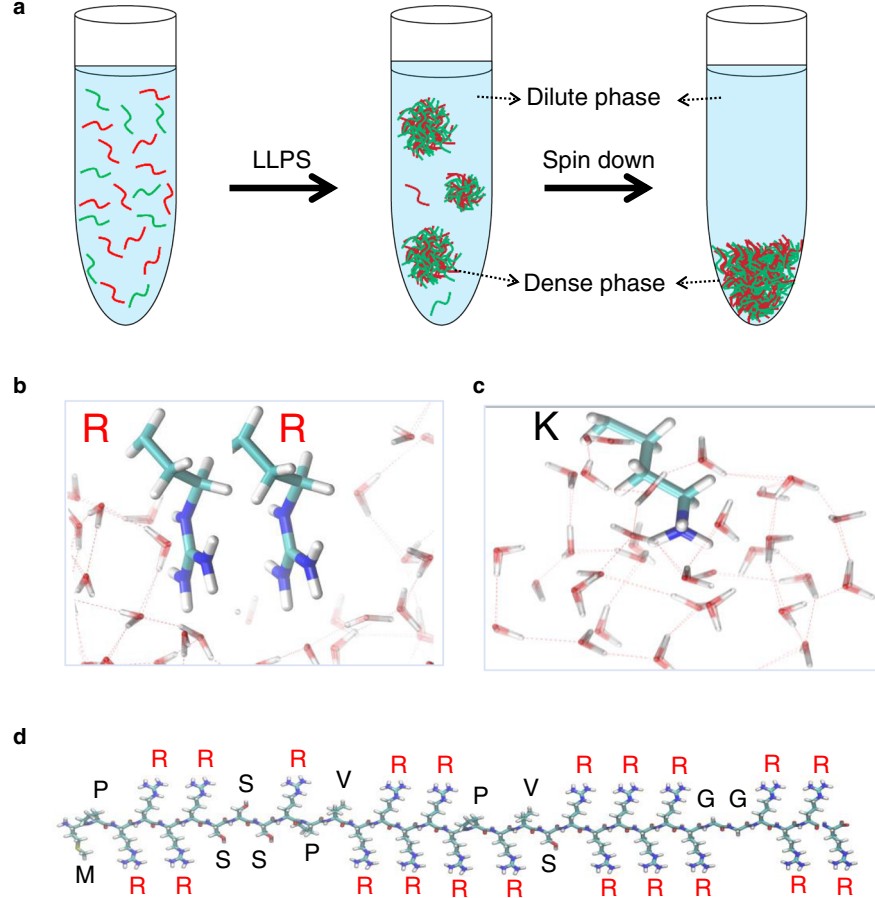

**Fig. 1 | Overviews of model systems. a** Illustrations of liquid-liquid phase separation (coacervation). **b** Chemical structure of arginine and **d**istributions of water around guanidinium group of arginine. **c** Chemical structure of lysine and distributions of water around ammonium group of lysine. **d** Primary amino acid sequence of protamine.

interact with proteins/RNAs to form phase-separated ribonucleoproteins (RNPs)[20–22]. Interestingly, arginine is a much more effective driver than lysine, despite both carrying the same single positive charge per residue[10,15–19]. For instance, in FUS, LAF-1, and Ddx4, substitutions of arginine-to-lysine has significantly reduced the phase-separation propensity, and vice versa, an increase in arginine content has been shown to lead to more favorable LLPS[10,15–19]. These observations have been rationalized to be due to arginine forming stronger cation−π interactions with proteins or RNAs containing aromatic rings than lysine[10,17,23,24]. Another recent study highlighted the role of arginine in stabilizing the coacervate phase by showing that poly-arginine peptides form condensates with much greater viscosity compared to poly-lysine (poly-Lys) peptides when mixed with polyU[25].

Along with the ability to form cation−π interactions, arginine stands apart from lysine in its ability to form π−π interactions. Indeed, arginine and lysine have different hybridization stemming from the guanidinium group in arginine and a primary amine in lysine (Fig. 1b, c). The guanidinium consists of three amino groups bound to a central carbon atom, hence described as a sp²-hybridized quasi-aromatic[26]. It acts as a hydrogen bond donor, interacting strongly with water only in the molecular plane of the guanidinium group, whereas the top and bottom surfaces of the guanidinium group tend to dehydrate readily (Fig. 1b)[27,28]. As a result, arginine has amphiphilic properties, allowing interactions by overcoming repulsive interactions between the positive charges of the stacked arginine[29–32]. The importance of π−π interactions in non-aromatic sp²-hybridized residues, including arginine, has been discussed by Forman-Kay, Chan, Stewart and by Knowles and coworkers, the latter in the context of LLPS at high salt

concentrations[18,32–34]. Further, this arginine-arginine pairing can concurrently occur with other interactions, including hydrogen bonding and electrostatic interactions[29–31]. In contrast, lysine has an ammonium group in its side chain that is a fully hydrated spherical ion (Fig. 1c)[29–31]. Hence, the NH₄⁺−NH₄⁺ pair is rather repulsive in water, making lysine-lysine stacking unfavorable.

Arginine can hence be viewed as having a dual nature, capable of forming both π−π and cation−π interactions[18,34]. It is however not clear from existing studies the extent to which the enhanced LLPS propensity of arginine-rich peptides over lysine-rich peptides arises from the colocalization of aromatic groups that engage in more favorable cation−π interactions with arginine than with lysine, or from the unique π−π interactions between arginines. Indeed, the majority of studies of arginine-rich polypeptides almost exclusively focus on complex coacervation with nucleotide-based counter anions to form LLPS and/or study the role of arginine in proteins with much greater sequence complexity and containing aromatic residues[17,25]. The goal of our study is to elucidate the competing roles of electrostatic interactions and hydrophobicity in driving LLPS using model systems purposefully stripped of any complexity. We focus on arginine-rich polypeptides and non-nucleotide-based counterions, all of which are free of aromatic residues, thus eliminating the influence of cation−π interactions from non-arginine residues, and allowing us to decouple the π−π interactions and the cation−π interactions between arginine and the aromatic group of nucleotides.

We choose salmon protamine as arginine-rich, and ε-poly-L-lysine (εPL) as lysine-rich, positively charged, protein models and hyaluronic acid as the negatively charged biopolymer to induce LLPS by complex

coacervation. This choice is further motivated by protamine and hyaluronic acids (HA) being both injectable biopolymers, already employed for biomedical applications, while protamine coacervation is considered a promising delivery mechanism for mRNA vaccines[35–39]. Protamine is a readily bio-available peptide with the highest arginine content among biological proteins. Protamine is mainly composed of arginine (~65 mol%), and contains methionine, serine, valine, proline, and glycine (no lysine, no anionic amino acids, and importantly, no aromatic amino acids) in the primary sequence (Fig. 1d). Using protamine as an arginine-rich system allows us to overcome the low solubility of poly-arginine, while protamine being a candidate material for drug delivery applications. We induced LLPS by complex coacervation between protamine and HA or between εPL and HA under charge matching conditions. To understand the interplay of electrostatic and hydrophobic interactions in governing LLPS behavior, we constructed two conditions: low salt condition in which arginine and lysine are positively charged, and high salt condition in which the positive charges of arginine and lysine are largely screened. Different salt types along the Hofmeister series were considered, in the context of self-coacervation in the absence of HA. We used an array of experimental techniques including microrheology, pulsed-field gradient nuclear magnetic resonance (PFG-NMR), electron paramagnetic resonance (EPR), and Overhauser dynamic nuclear polarization (ODNP), as well as computational tools relying on molecular dynamics and umbrella sampling simulations to characterize the conformational properties of the proteins under different salt conditions, and to quantify the hydration water dynamical and thermodynamical properties of arginine and lysine residues under LLPS conditions.

Our studies reveal dramatic differences in LLPS by complex coacervation as a function of salt concentration and temperature, with reentrant phase behavior observed for the arginine-rich, but not for the lysine-rich, peptide. Importantly, we demonstrate that the hydrophobicity of the arginine tunes the LLPS propensities at low and at high salt concentrations, as well as give rise to the emergent materials properties reflected in viscoelastic behavior of the dense LLPS phase. Our results can serve to guide the design of a phase-separating platform for drug and gene delivery systems controlled under conditions (modulated by salt concentration, crowding pressure and temperature) that give rise to reentrant LLPS.

## Results

### HA-Protamine and HA-εPL both form LLPS at low salt concentrations, but only HA-Protamine shows re-entrant behavior at high salt concentrations

Arginine-rich protamine, lysine-rich ε-poly-lysine (εPL) and hyaluronic acid (HA) were chosen as model constituents for investigating the effect of the guanidinium and ammonium side chains on the liquid-liquid phase separation (LLPS) of protein constituents. Protamine and εPL have similar molecular weights (~5 kDa). The LLPS of both HA-Protamine and HA-εPL coacervates is expected to be induced by complex coacervation mediated by multivalent electrostatic attraction of the oppositely charged side chains of the model polyelectrolytes. We examined the response of the LLPS of these model systems to ionic strength spanning 0 to 4 M in order to examine whether reentrant behavior is seen at high salt concentration, at which condition hydrophobic interactions should dominate. Under each condition examined, the yields of HA-Protamine and HA-εPL coacervates were determined by measuring the optical density of the suspension at 600 nm and the volume of the macroscopically separated dense phase (Fig. 2a, b). The yield of the complex coacervate phase decreased as ionic strength increased, until eventually the coacervates disappeared when [NaCl] reached 80 mM. This indicates that electrostatic attraction is one of the major driving forces of both polyelectrolyte systems at salt concentrations below 80 mM. Strikingly, HA-Protamine undergoes LLPS again above [NaCl] of 3 M, while the HA-εPL system does not

phase separate above 80 mM (Fig. 2a–c). The reentrant phase behavior of HA-Protamine coacervate suggests that the HA-Protamine coacervation is induced by hydrophobic as well as electrostatic interactions, while HA-εPL coacervation is mainly driven by electrostatic interactions.

Next, we examined the property of the complex coacervate in response to 1,6-hexanediol. Studies in the literature have shown that 1,6-hexanediol is an effective tool at disrupting LLPS held together by hydrophobic interactions[40,41]. When 10 wt% of 1,6-hexanediol was added, only HA-Protamine coacervate formed at high salt concentration was entirely dissolved (Fig. 2d, e). Hence, we propose that hydrophobic interaction is the major driving force for phase separation of HA-Protamine under high salt conditions.

To confirm the role of hydrophobic interaction in the LLPS of protamine, we investigated the effect of salt types along the Hofmeister series on the reentrant LLPS behavior in the absence of HA. As anticipated[34], salts earlier in the series known to induce "salting out" effects (e.g. KCl, NaCl) induce phase separation of protamine at lower salt concentration (2.75 M) by strengthening hydrophobic interactions. In contrast, salts later in the series known to induce "salting in" effects (e.g., RbCl, CsCl) drive phase separation of protamine at higher salt concentration (3.5 M and 4.5 M, respectively), as these salts tend to be less effective in strengthening hydrophobic interactions (Supplementary Fig. 1). These findings suggest that the reentrant LLPS of protamine at high salt concentration is driven by hydrophobic interactions between arginine residues.

### Strong intermolecular enthalpic interaction is a driver of high salt HA-Protamine coacervation

We next examined the phase behavior of HA-Protamine coacervation with temperature under different salt conditions to elucidate changes in the thermodynamic equilibrium in response to changes in ionic strength. At [NaCl] = 50 mM, the HA-Protamine coacervates show both upper critical solution temperature (UCST) and lower critical solution temperature (LCST) behavior, whereas the HA-Protamine coacervation at high salt concentration (here tested at [NaCl] = 3.5 M) exhibits only UCST behavior (Fig. 3a, b). The initial disruption of low salt HA-Protamine coacervates with increasing temperature (UCST) implies a dominant enthalpic contribution, and the reentrant phase separation of the same low salt HA-Protamine system at high temperature (LCST) reflects on the existence of a dominant entropic contribution in stabilizing the coacervate phase. Accordingly, the predominant UCST behavior observed for high salt HA-Protamine coacervation suggests that the enthalpic term driven by strong molecular interaction forces, presumably between arginine moieties, are dominant over the entropic term. In contrast, HA-εPL exclusively shows LCST behavior at [NaCl] = 50 mM, as discussed in previous studies[3], indicating that entropy gain, likely driven by hydration water release, is a main driver for HA-εPL complex coacervation (Fig. 3c). The observation that HA-εPL does not undergo phase separation at high salt concentrations in the examined temperature range suggests that strong intermolecular enthalpic interactions are not the dominant drivers of LLPS (Fig. 3d). Commonly, electrostatic interactions are accompanied by counterion release and hydration water release upon coacervation, giving rise to entropic contributions that drive LCST behavior[3,42]. Concurrently, the hydrophobic self-interaction of arginine molecules through π–π stacking of guanidinium side chain is at play that is known to decrease with temperature, giving rise to enthalpic contribution that drive UCST behavior[43]. Thus, HA-Protamine coacervation at low salt concentrations, driven by the action of both hydrophobic and electrostatic interactions, displays dual UCST and LCST behavior. In contrast, for HA-Protamine coacervation at high salt concentrations, it shows only UCST behavior because electrostatic interactions are screened out and hence entropic contribution to coacervation is insignificant. In the absence of salt, the number of hydrogen bonds between the hydration

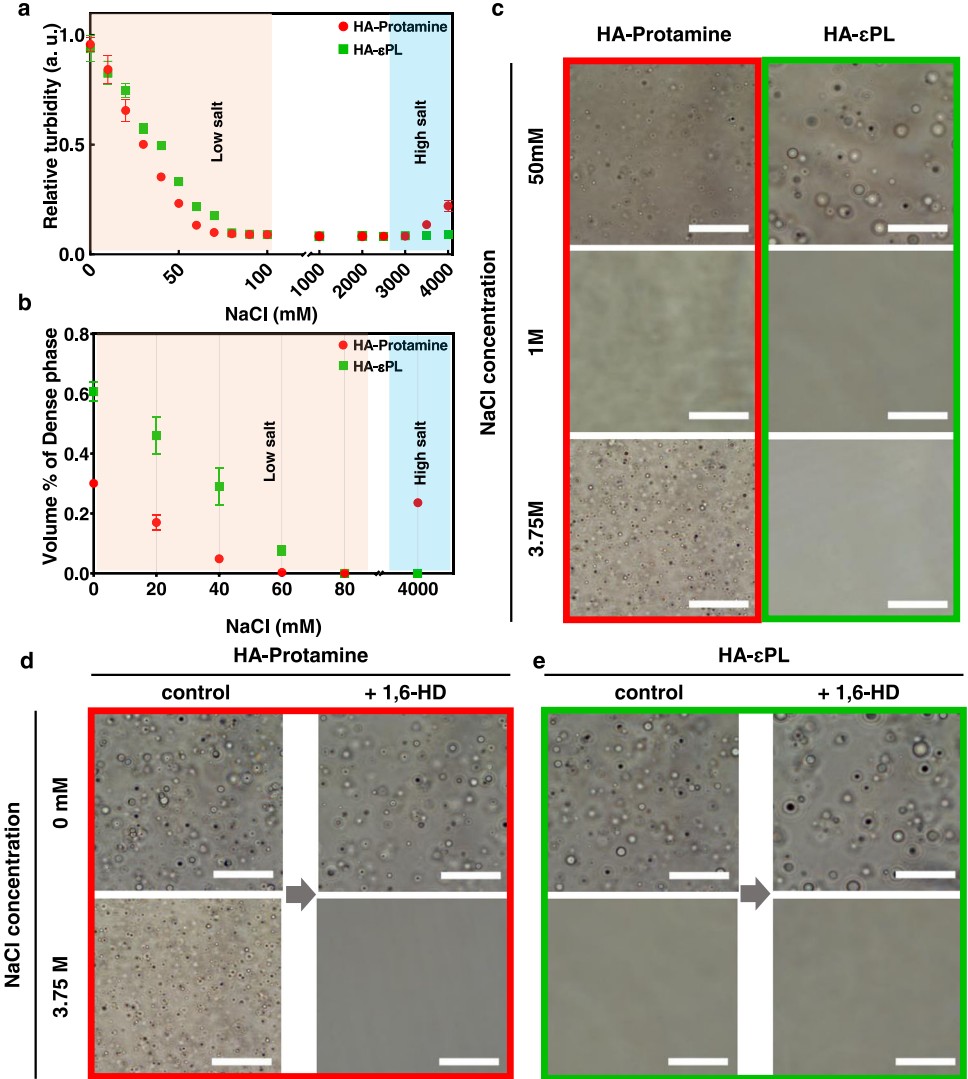

**Fig. 2 | Different phase behaviors under high salt conditions. a** Relative turbidity of HA-Protamine and HA-εPL at 600 nm at specific NaCl concentration. $n = 3$ independent experiments. Data are presented as mean values ± SD. a.u. arbitrary units. **b** Volume of dense phase divided by total solution volume at specific NaCl concentration. $n = 3$ independent experiments. Data are presented as mean values ± SD. **c** Representative images of HA-protamine and HA-εPL condensates in the presence of increasing of NaCl. All measurements were performed in triplicate at room temperature. **d** HA-Protamine upon addition of 10 wt % 1,6-Hexanediol at certain NaCl concentrations. **e** HA-εPL condensates upon addition of 10 wt % 1,6-Hexanediol at certain NaCl conditions. All experiments were performed in triplicate at room temperature Scale bar is 40 µm. The data are available in the Figshare repository under [https://doi.org/10.6084/m9.figshare.21509343.v3].

water and protamine is much greater than in the case of poly-Lys (Supplementary Fig. 2), implying that the release of hydration water upon coacervation of HA-Protamine can much more effectively contribute to entropy-driven LCST behavior at low salt concentration. In contrast, at high salt concentration, the number of hydrogen bonds between hydration water and the polymer is comparable between protamine and poly-Lys. Instead, at high salt concentration the dominant effect that drives HA-Protamine UCST-LLPS is the enthalpic interaction among Arg residues. This is evidenced by the major increase of intramolecular contacts of Arg in the protamine chain upon increasing the salt concentration (Supplementary Fig. 7b, d).

## Arg in protamine is more hydrophobic than Lys in poly-lysine at the residue level

To understand the molecular origin of hydrophobic interactions between Arg groups in water, we need to determine the hydration properties of Arg in protamine, and contrast them to those of Lys in poly-Lys. The hydrophobicity of an amino acid in protein can be affected by the residue-water hydrogen bonding and the topography

of the protein surface which can alter the hydration water network. In fact, IDPs feature continuous conformational transition, and the water thermodynamics within the hydration shell of a particular amino acid residue can be affected by nearby residues. Therefore, hydrophobicity of amino acids is a context-dependent quantity. Hence, we aim to determine the molecular hydrophobicity of Arg and Lys in different sites along the sequence of protamine and poly-Lys, respectively. We use indirect umbrella sampling (INDUS) simulations[44,45] to characterize the residues hydrophobicity in all atom representation in explicit solvent, using the amber03ws force field[46,47]. We compute the dehydration free energy of Arg and Lys in protamine and poly-Lys by employing an unfavorable harmonic biasing potential to sample over the range of water densities (by expelling or dragging water) in the dynamical hydration volume of the target residue. The dynamical hydration volume of each residue under study (Lys and Arg) is defined as the union of pinned spherical sub-volumes with radius $R_v$ to the center of every heavy atom (carbon, oxygen, nitrogen, and sulfur) of the amino acid, whose position can evolve during MD simulation; we include only the first hydration shell waters in the free energy

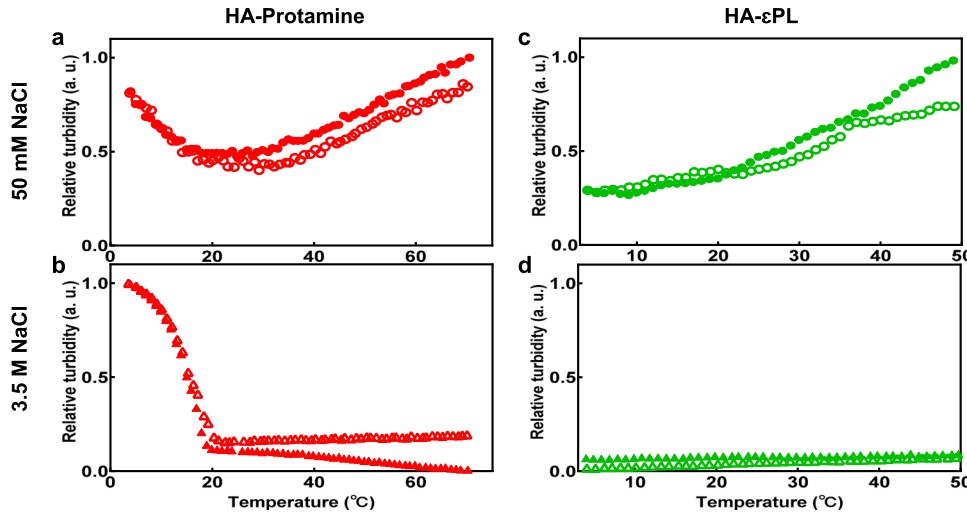

**Fig. 3 | Phase behavior of each system depending on the temperature.**
**a** HA-Protamine coacervate in additional 50 mM and **b** 3.5 M NaCl. **c** HA-εPL coacervate in additional 50 mM NaCl and **d** 3.5 M NaCl. Turbidity was measured at 600 nm over temperature by cooling cycle (filled) and heating cycle (empty). a.u. arbitrary units. The data are available in the Figshare repository under [https://doi.org/10.6084/m9.figshare.21509343.v3].

calculation by choosing $R_v = 0.55$ nm. The unbiased free energy from INDUS is then obtained by using the Unbinned Weighted Histogram Analysis Method (UWHAM)[48,49]. Representative MD snapshots of poly-Lys and protamine are shown in Fig. 4a. In a poly-Lys chain of 33 lysine residues, the dehydration free energy of Lys is independently calculated in the hydration volumes of $Lys_2$, $Lys_{16}$ and $Lys_{33}$. Similarly in the protamine chain, the dehydration free energy of Arg is obtained in the hydration volumes of $Arg_4$, $Arg_{16}$ and $Arg_{32}$, where the subscript indicates the residue index in sequence (Supplementary Fig. 3). The calculated free energies were used to compute the average dehydration free energy per water molecule in the hydration volume of Lys and Arg, shown in Fig. 4b. A more hydrophobic surface will cost less dewetting free energy per water molecule. The results reveal that the free energy of expelling a water molecule from the hydration shell of Arg is by ~0.6$k_B$T lower than that of Lys. Hence, Arg in protamine is more hydrophobic than Lys in poly-Lys at the residue level. As illustrated in Supplementary Fig. 3b, the dewetting free energy of Arg in the protamine chain can vary along the sequence, depending on the type of the nearby residues. This nonidentical dewetting free energy behavior indicates the occurrence of context-dependent hydrophobicity of a particular residue in a peptide featuring a nonuniform chemical sequence. However, the Arg dewetting free energy remains unchanged in a homopeptide poly-Arg chain (Supplementary Fig. 3c) that mainly features expanded conformations with minimal structural complexity. The average dewetting free energy of Arg in protamine and poly-Arg is identical (Supplementary Fig. 3d). This indicates that while the local hydrophobicity of a residue can be different along a complex peptide, its globally averaged dewetting free energy can still be similar to that of the residue in a homopeptide.

We experimentally determined the net biopolymer concentration by quantitative amino acid analysis in the dense phase of HA-Protamine formed under low salt and high salt (4 M NaCl) conditions, which is a proxy for inducing dehydration conditions (Fig. 4c and d). The results show that both the protamine and hyaluronic acid densities are greater in the coacervate phase of high salt HA-Protamine compared to low salt HA-Protamine. Interestingly, the ratio of protamine ($158.3 \pm 20.7$ mg/mL) to hyaluronic acid ($165.2 \pm 22.8$ mg/mL) is ~1 in the low salt HA-Protamine coacervate phase, whereas the ratio of protamine ($1227.8 \pm 76.9$ mg/mL) to hyaluronic acid ($680.9 \pm 42.3$ mg/mL) is ~1.8 in the high salt HA-Protamine coacervate phase. These results show higher packing of protamine with each other relative to HA,

suggesting that there are stronger hydrophobic self-interactions of protamine molecules, presumably via direct guanidinium π−π stacking of Arg residues[29–31].

## Hydrophobic interactions among arginine dictate physical properties of the dense coacervate phase

To investigate the effect of hydrophobic interactions on the physical properties of the coacervate phase, we compared the polyelectrolyte concentration, viscosity, and interfacial tension of low salt HA-εPL, low salt HA-Protamine coacervate phase without additional NaCl, and high salt HA-Protamine coacervate phase formed at high salt ([NaCl] = 4 M) condition (Table 1). The low salt HA-Protamine coacervate phase (~323 mg/mL) is around 1.7 times more concentrated than the low salt HA-εPL coacervate phase (~188 mg/mL). As previously shown, the coacervate phase of high salt HA-Protamine (~1909 mg/mL) is about 6 times more concentrated than of low salt HA-Protamine (~323 mg/mL). Similarly, the viscosity of the low salt HA-Protamine coacervate phase (0.3052 Pa s) is approximately an order of magnitude greater than that of the low salt HA-εPL coacervate phase (0.0244 Pa s), while the interfacial tension of the low salt HA-Protamine coacervates ($3.19 \times 10^{-5}$ N/m) 1.4 times higher than of low salt HA-εPL ($2.34 \times 10^{-5}$ N/m). The viscosity (28.5 Pa·s) and interfacial tension ($4.20 \times 10^{-3}$ N/m) of the high salt HA-Protamine coacervate phase are two orders of magnitude greater compared to that of the low salt HA-Protamine and the HA-εPL coacervate phase. However, high salt HA-Protamine coacervates are still considered to have liquid-like properties as shown in Supplementary Fig. 4. Higher viscosity and higher interfacial tension correlate with stronger enthalpic intermolecular interactions. Hence, these observations reinforce our earlier stated hypothesis that stronger attractive intermolecular interactions among the constituent hold together the HA-Protamine, compared to the HA-εPL, coacervate phase.

The next question we ask is whether the HA-protamine coacervate phase display viscoelastic, i.e. materials, properties. To do so, we determined the storage modulus (G′) and loss modulus (G″) of low salt and high salt HA-Protamine coacervates (Fig. 5). The loss modulus is greater than the storage modulus at low salt concentration over the frequency range 0.02 ~ 7.38 s⁻¹. Thus, low salt HA-Protamine coacervates predominantly exhibit viscous, liquid behavior. In contrast, high salt HA-Protamine coacervates exhibit viscoelastic fluid behavior with a crossover frequency (G′ value equals G″ value) at lower frequencies compared to low salt HA-Protamine coacervates. These

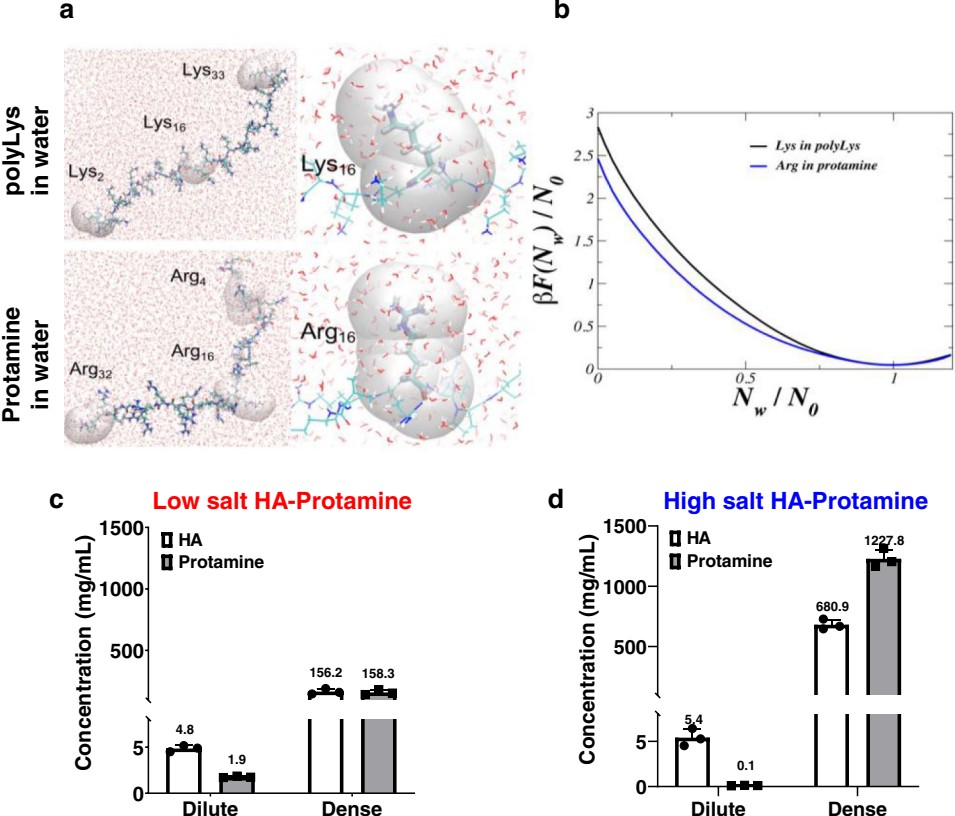

**Fig. 4 | Dehydration free energy calculation from INDUS and polymer concentrations in HA-Protamine coacervates. a** The representative MD snapshots of poly-lysine and protamine are illustrated. The probe volumes in which the dehydration free energies are independently calculated for $Lys_2$, $Lys_{16}$ and $Lys_{33}$ of polylysine and $Arg_4$, $Arg_{16}$ and $Arg_{33}$ of protamine is shown in transparent-gray. **b** The average dehydration free energy per water molecule in the hydration volume of Lys and Arg obtained from INDUS simulations is shown; $N_w$ is the instantaneous number of water molecules in the hydration volume and $N_0$ is the number of water molecules in the hydration shell of the residue at equilibrium. **c** Concentrations of

hyaluronic acid and protamine within dilute and dense phase without additional salt. **d** Concentrations of hyaluronic acid and protamine within dilute and dense phase with 4 M NaCl. Measurements were conducted by quantitative amino acid analysis at RT and all measurements were performed in triplicate except for the dense phase of high salt HA-Protamine. The dense phase concentration of high salt HA-Protamine was calculated from volume of each phase and dilute phase concentration. All experiments were conducted in triplicate independently. Data are presented as mean values ± SD. The data are available in the Figshare repository under [https://doi.org/10.6084/m9.figshare.21509343.v3].

observations show that inter- and intra- molecular hydrophobic interactions between protamine in the HA-Protamine coacervate phase not only render the coacervate phase denser, but also impart viscoelastic property to the coacervate phase. Changes in the viscoelastic properties of the coacervate are expected to play important roles in regulating the mechanical response (fragile or resilient) of the dense coacervate phase against stress induced by the chromatin constituents, which in turn will affect many cellular processes, such as, signaling and gene regulation.

## Protein chain dynamics in HA-Protamine and HA-εPL coacervates

We assessed the mobility of proteins in the coacervate phase by performing fluorescence recovery after photobleaching (FRAP) of the entire droplet. The results show that in both HA-Protamine and HA-εPL coacervates, protamine and εPL molecules can be exchanged with the surrounding protamine and εPL molecules in the dilute phase (Fig. 6a, top and middle). However, at the high salt concentration, the exchange of protamine molecules between the dilute and dense phases was greatly reduced (Fig. 6a, bottom). Next, the rotational dynamics of the polymer molecules was evaluated by continuous-wave Electron Paramagnetic Resonance (cw-EPR) lineshape analysis of spin labels tethered to the biopolymer surface. To achieve spin labeling, the amine groups of protamine and εPL were functionalized with the nitroxide radical, here 4-carboxy TEMPO. Consequently, the spin labels are

statistically distributed throughout the εPL chain, but are located exclusively at the N-terminal of protamine (Supplementary Fig. 5). The spin-labeled SL-protamine or SL-εPL were mixed with hyaluronic acid (HA), and upon LLPS, centrifuged down to isolate the dilute and dense phase. The rotational correlation time ($t_{corr}$) was derived from cw-EPR lineshape analysis assuming isotropic rotational motion of the spin label. On SL-εPL in the HA-εPL dense phase $t_{corr}$ was found to be 454 ps, representing dynamic liquid state behavior (Fig. 6b top). Consistent with the FRAP results, the motion of SL-protamine in low salt HA-Protamine coacervate phase was found to be also consistent with liquid state behavior, as represented with $t_{corr}$ = 151 ps (Fig. 6b middle). The $t_{corr}$ value is lower (more dynamic) for SL-protamine despite the higher polyelectrolyte concentration in the HA-Protamine, compared to the HA-εPL, coacervate phase. The absolute value for $t_{corr}$ of SL-protamine may be lower than for SL-εPL because the spin label of SL-protamine is exclusively tethered to the protamine chain ends that are more dynamic than the middle of a biopolymer chain. Overall, SL-protamine and SL-εPL are highly mobile, representing behavior of a liquid coacervate phase formed under low salt conditions. In contrast, the rotational motion of SL-protamine in high salt HA-Protamine coacervates is significantly hindered. This is reflected in the cw-EPR lineshape whose major population (a minor fast motion population is present, likely originating from residual dilute phase) presents a spectral feature in the "rigid limit" for the spin labels of SL-protamine, despite their positions at protamine chain ends (Fig. 6b bottom). This

**Table 1 | Physical properties of the dense phase**

| System | Concentration (mg/mL) | Viscosity (Pa s) | Interfacial tension (N m$^{-1}$) | Ref. |
|---|---|---|---|---|
| Low salt HA-εPL | 188 | 0.0244 | 2.34 × 10$^{-5}$ | 3 |
| Low salt HA-Protamine | 323.4 | 0.3052 | 3.19 × 10$^{-5}$ | This study |
| High salt HA-Protamine | ~ 1909 | 28.47 | 4.20 × 10$^{-3}$ | This study |

The values of HA-εPL coacervate are from ref. 3. The physical properties of HA-Protamine coacervate were measured in this study. The concentrations are the total polyelectrolyte concentration within the dense phase calculated in Fig. 4. All experiments were conducted at room temperature.

observation is consistent with the dramatically higher viscosity and viscoelastic property of the dense HA-Protamine coacervate phase formed under high salt conditions. These results further validate that enhanced hydrophobic interactions from arginine impart material properties to arginine-rich phases of LLPS (Supplementary Fig. 6).

The conformational features of the peptide, governed by intra-molecular interactions, can influence collective behaviors of peptides, such as their aggregation and LLPS properties. For example, strong intramolecular interactions that lead a single peptide to adopt more collapsed globular states, can, in the presence of other peptides, also facilitate intermolecular interactions that drive LLPS[50]. To determine the conformational properties of poly-Lys and protamine chains and gain molecular insight into their collective behavior, we performed atomistic molecular dynamics (MD) simulations in explicit solvent. To do so, we study the conformational landscape of poly-Lys and prota-mine by monitoring the radius of gyration ($R_g$) and the end-to-end distance ($d_{ee}$) of the chains in a large ensemble of structures obtained from 600 ns long MD simulations in the NPT ensemble at $T = 300$ K and $P = 1$ bar. Each chain was simulated without salt and in the presence of 3.5 M [NaCl] (high salt). As shown in Fig. 6c, the MD simulations without salt feature by a shallow probability distribution of $R_g$ and $d_{ee}$ indicating that the poly-Lys is mainly in extended conformational states with rare intramolecular contacts between the nonadjacent residues along the sequence (Supplementary Fig. 7). However, at high salt concentration, poly-Lys features less extended conformations, presumably due to screening of the repulsive electrostatic interaction among residues by the excess salt, see Fig. 6d. Interestingly, as illustrated in Fig. 6e, protamine features a main minima corresponding to extended conformations, and a more shallow one corresponding to collapsed conformations, indicating an interplay between repulsive electrostatic interactions (among Arg residues that promotes occurrence of extended conformations) and hydrophobic interactions (that tend to drive the chain toward a more collapsed globular state). The effect of Arg hydrophobicity is partially reflected in a smaller average $d_{ee}$ and $R_g$ of protamine than poly-Lys, by 0.57 nm and 0.38 nm, respectively. These competing electrostatic and hydrophobic inter-actions among Arg residues become even more evident in the high salt concentration regime, Fig. 6f—here, the protamine population is skewed towards the collapsed conformations, as evidenced by enhanced population at a smaller average $d_{ee}$ and $R_g$ of protamine (to be contrasted with the values of poly-Lys at 0.93 nm and 0.73 nm, respectively). At high salt concentration, the electrostatic interactions are screened, and the lower net charge of protamine than poly-Lys suggests greater conformational fluctuations of protamine that can lead to effective association with other chains. This conformational property of protamine is similar to that of poly-Arg (Supplementary Fig. 8 right). In addition, we check the effect of the heterotypic nature of protamine on its conformational behavior by computationally comparing the collective behavior of a pseudo-protamine-RtoK chain, in which all the Arg residues of protamine are substituted by Lys with the poly-Lys chain. We observed that the $d_{ee}$ and $R_g$ of the pseudo-protamine-RtoK chain are slightly lower than the poly-Lys chain, however, the enthalpic intramolecular interactions of the pseudo-protamine-RtoK (evidenced by the contact map) are mostly similar to that of the poly-Lys chain (Supplementary Fig. 8 left)

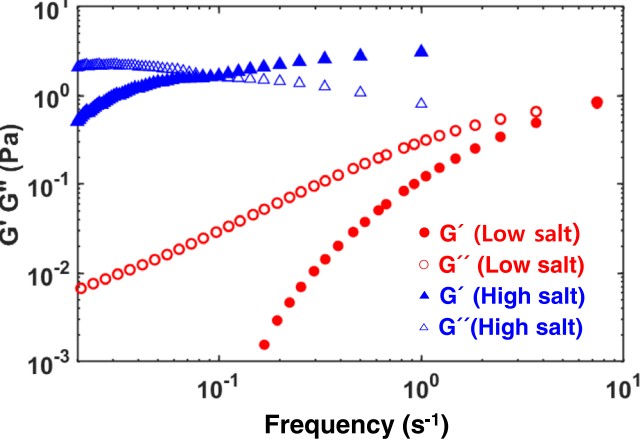

**Fig. 5 | Viscoelasticity of HA-Protamine coacervates under low and high salt conditions.** G′ (Elastic modulus) and G″ (Viscous modulus) of low salt and high salt HA-Protamine coacervates measured at room temperature. The data are available in the Figshare repository under [https://doi.org/10.6084/m9.figshare.21509343.v3].

## Dynamics of interstitial and hydration water in HA-Protamine and HA-εPL coacervates

The dynamic and structural properties of water interacting with the proteins are considered crucial for the structure, function, and free energetic properties of proteins;[51,52] We hence characterized the dynamical property of water interacting with the protein surface. We performed pulsed-field gradient nuclear magnetic resonance (PFG-NMR) that measures the molecular diffusivity of water by tracking displacement of water over micrometer distances, and Overhauser DNP (ODNP) that captures instantaneous movement of water with sub-nanosecond range correlation times across nanometer distances, again yielding molecular diffusivity of water[53,54]. If ODNP is performed using spin labels tethered to the protamine chain ends or statistically labeled at the εPL side chains (Supplementary Fig. 5), ODNP is sensitive to hydration water dynamics within 0.5-1.5 nm of the spin-labeled biopolymer surface.

First, phase-separated dilute and dense coacervate phase of the HA-Protamine and HA-εPL coacervate system were prepared by centrifugation, and the self-diffusion coefficients of water in these phases determined by PFG-NMR. In the low salt HA-εPL coacervate system, the interstitial water diffusivity is found to be 2.35 × 10$^{-9}$ m$^2$/s in the dilute phase, corresponding to bulk water diffusivity values at the given sample temperature, and 1.51 × 10$^{-9}$ m$^2$/s in the dense coacervate phase (Fig. 7a), consistent with a recent report on HA-εPL complex coacervates[3]. In the low salt HA-Protamine coacervate system, the interstitial water diffusivity in the dilute and dense phase is found to be 2.52 × 10$^{-9}$ m$^2$/s and 1.34 × 10$^{-9}$ m$^2$/s, respectively (Fig. 7b). In both systems, the dif-fusivity of water within the dilute phase did not decrease com-pared to that (2.52 × 10$^{-9}$ m$^2$/s) of deionized water (DW), while the reduced value of the diffusion coefficient of water in the dense

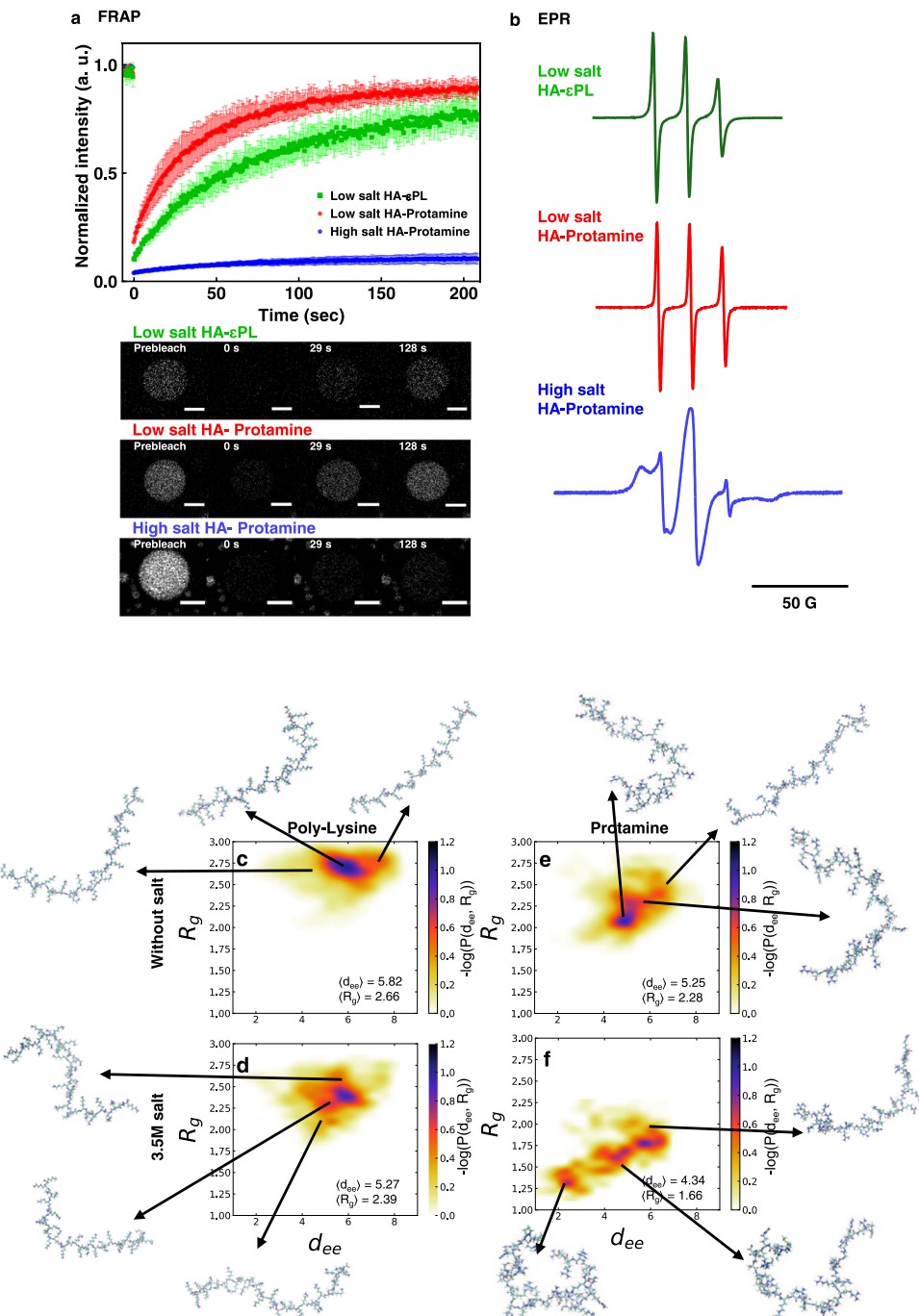

**Fig. 6 | Chain dynamics and conformational properties of protamine and εPL.**
**a** FRAP curve of entire droplet of HA-Protamine ($n = 4$) and HA-εPL complex coacervation ($n = 3$) without additional salts, and HA-Protamine complex coacervation under high salt condition ($n = 4$). (top). Representative images (bottom). Scale bar = 10 μm. Data are presented as mean values ± SD. a.u. arbitrary units. **b** EPR spectra of spin-labeled εPL and protamine in the dense phase of HA-εPL complex coacervation (top), HA-Protamine complex coacervation (middle) without additional salts,

and HA-Protamine complex coacervation under high salt condition (bottom).
**c–f** Determining the proteins conformational properties from MD simulations.
**c** and **e** represent the probability distribution of the radius of gyration ($R_g$) and the end-to-end ($d_{ee}$) distance of εPL and protamine chains without salt, respectively. In **d** and **f**, the probability distribution of $R_g$ and $d_{ee}$ are calculated in presence of 3.5 M salt in solution. The data are available in the Figshare repository under [https://doi.org/10.6084/m9.figshare.21509343.v3].

phase is consistent with the higher viscosity of the dense phase. Overall, the interstitial water molecules diffuse relatively unhindered compared to the diffusion of the polymer constituents in the highly concentrated dense phase in both systems. Interestingly, the self-diffusion coefficient of interstitial water in the dense phase of HA-Protamine ($1.34 \times 10^{-9}$ m²/s) is similar to that in the dense phase of HA-εPL ($1.52 \times 10^{-9}$ m²/s), despite the 1.7 times higher polyelectrolyte concentration and an order of magnitude

higher viscosity in the dense HA-protamine coacervate, than the HA-εPL coacervate, phase.

ODNP can capture two types of dynamical properties of water, complementary to PFG-NMR: the diffusion coefficient of average hydration water ($D_{\text{coupling factor}}$) that includes the effect of bound and freely exchanging water with nanosecond scale correlation times, and the diffusion coefficient of freely exchanging water ($D_{\text{ksigma}}$) with picosecond scale correlation times, as discussed in previous

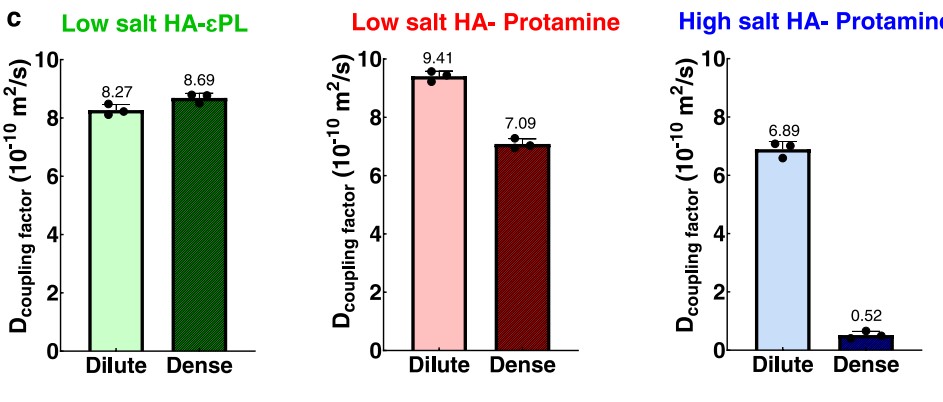

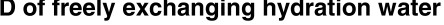

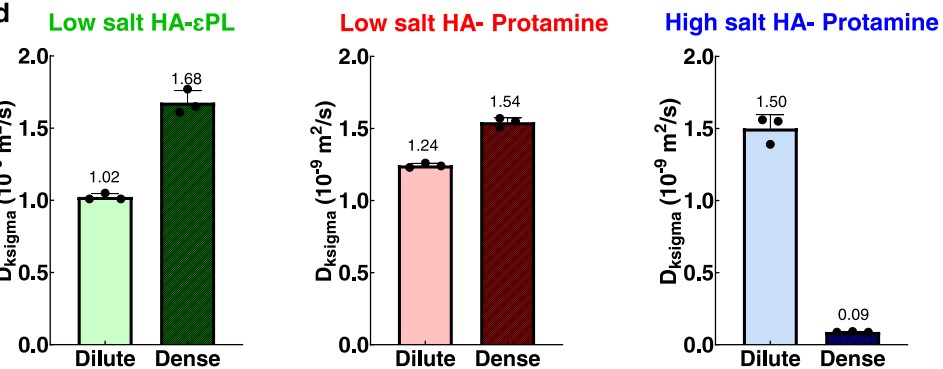

**Fig. 7 | Interstitial water and hydration water dynamics. a** Measurement of the diffusion coefficient of interstitial water within the dilute and dense phase of HA-εPL complex coacervation using PFG-NMR. Reproduced from ref. 3. **b** Measurement of the diffusion coefficient of interstitial water within the dilute and dense phase of HA-Protamine complex coacervation ($n = 3$) and DW ($n = 1$) using PFG-NMR. **c** $D_{coupling\ factor}$ of low salt HA-εPL (left), low salt HA-Protamine (middle), and high salt HA-Protamine (right) coacervation measured via ODNP. **d** $D_{ksigma}$ values were obtained via ODNP for low salt HA-εPL (left), low salt HA-Protamine (middle), and high salt HA-Protamine (right) coacervation. For all, $n = 3$ independent experiments. All measurements are conducted at room temperature. Data are presented as mean values ± SD. The data are available in the Figshare repository under [https://doi.org/10.6084/m9.figshare.21509343.v3].

studies[54,55]. The results show that $D_{coupling\ factor}$ within the dense phase of low salt HA-εPL ($8.69 \times 10^{-10}$ m²/s) is slightly faster than that in the dilute phase ($8.27 \times 10^{-10}$ m²/s), as reported in Ref. 3 (Fig. 7c left). In contrast, $D_{coupling\ factor}$ in the dense phase of the low salt HA-Protamine ($7.09 \times 10^{-10}$ m²/s) is by a factor of ~1.3 slower than in the dilute phase ($9.41 \times 10^{-10}$ m²/s) (Fig. 7c middle). Furthermore, the water diffusivity as measured by $D_{coupling\ factor}$ in the low salt HA-Protamine dense coacervate phase is slower than in the low salt HA-εPL dense coacervate phase, which is expected given the higher concentration and viscosity in the HA-Protamine dense coacervate phase. Surprisingly, water is overall highly mobile and freely diffusing, even in the highly concentrated dense phase of low salt HA-Protamine. The diffusivity of

water measured on the surface of spin-labeled biopolymers, as reflected in $D_{coupling\ factor}$, is about a factor of 2-3 times lower than the interstitial diffusivity measured by PFG-NMR. This is an expected retardation factor due to geometric effects of the biopolymer surface (because diffusion occurs away, not into, the surface) and due to favorable interaction between the biopolymer and water that may be stronger than water-water hydrogen bond interactions. Still, the observed dynamics of average hydration water according to ODNP is consistent with that measured by PFG-NMR, in that the interstitial water moves freely in both low salt HA-εPL and low salt HA-Protamine coacervation. In contrast, the mobility of average hydration water is an order of magnitude slower in the dense phase of high salt

HA-Protamine, with $D_{coupling\ factor}$ ~ $0.52 \times 10^{-10}$ m²/s (Fig. 7c right). In other words, the polymer and water constituents are tightly locked together and their movements dramatically hindered in the high salt HA-Protamine dense coacervate phase.

Next, we evaluate the freely exchanging water dynamics ($D_{ksigma}$) with picosecond range correlation times. Under low salt conditions, the diffusivity of water in the dense phase of HA-εPL ($1.68 \times 10^{-10}$ m²/s) and HA-Protamine ($1.54 \times 10^{-10}$ m²/s) systems is faster than in the dilute phase of HA-εPL ($1.02 \times 10^{-10}$ m²/s) and HA-Protamine ($1.24 \times 10^{-10}$ m²/s) systems, respectively (Fig. 7d left and middle). It seems that charge neutralization by complexation of oppositely charged polyelectrolytes contributes to less hindered diffusion of water near the surface of charge-neutralized biopolymers, presumably due to weakened biopolymer-surface interactions. In contrast, the surface water diffusivity is again dramatically slowed down in the high salt HA-Protamine dense phase ($0.09 \times 10^{-10}$ m²/s) compared to the low salt coacervate phase, suggesting that even surface water is highly immobilized in the high salt HA-Protamine system (Fig. 7d right). We can conclude that interstitial and surface hydration water is freely diffusing within the dense coacervate phase of low salt HA-εPL and low salt HA-Protamine, but immobilized within the dense coacervate phase of high salt HA-Protamine coacervate phase that displays viscoelastic materials property.

## Discussion

Arginine-rich peptides or proteins are an important class of biomolecules that have applications ranging from carriers for genes, drug, and even adhesive molecules, due to their cationic nature, nucleic acid binding ability, and cell-penetrating properties, as well as biocompatibility. These proteins have the ability to undergo LLPS, and the resulting protein droplets have controllable cargo recruitment and release properties in response to external triggers, making them one of the most promising development options for delivery platforms[12–14]. Arginine is a much more potent inducer of LLPS than other cationic amino acids carrying the same charge (such as lysine), suggesting that electrostatic interactions alone do not fully account for LLPS propensity of cationic polymers. We investigated two model cationic-rich systems that could undergo LLPS: poly-lysine and protamine, the latter being an arginine-rich, clinically relevant, highly biocompatible and injectable biomolecules used in cardiac surgery to neutralize the anticoagulant effects of heparin[35,36]. We find that both protamine and poly-lysine undergo LLPS at low salt concentration, but that only protamine exhibits salt and temperature-dependent reentrant phase behavior. Furthermore, we show that the viscoelastic physical properties of the protamine and poly-lysine coacervate phases differ. Our combined experimental and computational studies reveal that the reentrant LLPS behavior of protamine at high salt concentrations and the more viscous nature of the protamine-rich coacervate phase compared to that of poly-lysine, is due to the greater hydrophobicity of arginine over lysine, in particular, its ability to form π−π stacking interactions. The use of very high salt concentration in this study well exceeds physiological ionic strength, but is serving as a tool to study hydrophobic properties of arginine. High salt conditions mimic dehydrating conditions frequently found in the crowded cellular environment. Similar dehydration condition as modeled in this study can be achieved by using molecular crowders or by using high concentrations of multi-charged metabolites found under cellular conditions[3,56,57]. Given the abundance of multi-charged metabolites and the fact that their levels are altered by cellular conditions, such as stress and cell division, the high salt concentration and the reentrant phase behavior in this study are expected to account for cellular protein phase separation under cellular condition with enhanced hydrophobic interactions.

The ability of protamine to undergo LLPS with HA at low salt concentrations, dissolves with increasing salt concentration and then shows reentrant LLPS formation at high salt concentrations, in a salt type and temperature-dependent fashion showcases the potential application of arginine-rich systems for salt triggered cargo (a nucleic acid or drug) uptake and release (Fig. 2). Additionally, HA-Protamine shows temperature-dependent phase behaviors, which again depend on the ionic strength (Fig. 3). At low salt concentrations they display UCST and LCST behavior, while at high salt concentrations they only show UCST behavior. Assuming HA as a cargo, our results show that protamine can be used as a phase-separating delivery vehicle that can control the recruitment and release of cargo in response to ionic strength, salt type and temperature. We hence expect that this study will provide insight into designing and formulating phase-separating arginine-rich domain-based delivery systems with appropriate material properties.

Understanding the role of arginine in tuning the physical properties of the dense LLPS phase is also highly relevant for developing underwater adhesives. Coacervates composed of mfp-3 and mfp-5, key adhesive primer proteins for the mussel underwater adhesion, are rich in arginine, and not in lysine, unlike other mussel foot proteins (e.g., mfp-1)[58,59]. At physiological salt concentrations in the mussel foot, mfp-3 coacervates would be fluidic to facilitate secretion and delivery to the substrate. Then, the high ionic strength of the seawater would induce the hardening of mfp-3 coacervates, which concentrates the adhesive proteins, imposes viscoelasticity and promotes a more robust adhesion at the plaque-substrate interface. Our results suggest that arginine might be the most potent mechanisms for regulating the physical properties of mfp-3 coacervates in this series of processes. Specifically, we demonstrated that low salt HA-Protamine coacervates exhibit viscous and highly dynamic liquid-like character with low interfacial tension, while high salt HA-Protamine coacervates driven by hydrophobic interactions exhibit viscoelastic and immobile solid-like character with high interfacial tension. Importantly, the prominent role of arginine in giving rise to high viscosity and viscoelastic properties of the dense coacervate phase, as well as salt-dependent reentrant behavior and temperature induced phase transitions through hydrophobic interactions could all be recapitulated using purposefully simplified model systems consisting of protamine and HA. It should be possible to use other arginine-rich biopolymers with additional bioactive properties for the design and improvement of biological adhesion, as well as for mimicking the biological wet-adhesion for developing surgical glues, or antifouling surface.

Last, but not least, arginine is also prevalent in RNA-binding proteins (RBPs), which are one of the main components of phase-separated stress granule (SGs) (Supplementary Fig. 9)[20–22]. Under physiological conditions, SGs have been understood to be involved in storing untranslated mRNAs, resulting in translational arrest[60]. Based on previous observations, SG-related RBPs with disordered arginine-rich motifs undergo phase separation in vitro and in vivo[10,11,61,62]. However, SGs induced and co-localized by arginine-rich peptides display reduced dynamic properties[21,63] and sometimes, gelation or aggregation of SGs is known to be associated with neurodegenerative diseases[60]. In agreement with this, our results show that the high salt HA-Protamine coacervates driven by hydrophobic interactions have solid-like properties of elasticity and high immobility of protamine and water, along with an increase of protamine concentration within the coacervate phase. Additionally, at physiological low salt concentrations, arginine-rich protein coacervates are much denser and exhibit viscoelastic properties, but still contain dynamic protein chains and water comparable to lysine-rich protein coacervates. This appears to indicate that arginine-rich domains/proteins of SGs might function to retain and protect mRNA by generating tight binding and viscoelastic environment for mRNA. We hence speculate that the activity and function of biomolecular constituents can be maintained in engineered arginine-rich membraneless organelles or biomolecular condensates.

## Methods

### Materials

Protamine sulfate salt from salmon (5.1 kDa, p4020, Lot# SLBW4512, Oncorhynchus keta), Poly(ethylene glycol) (10 kDa), Sigmacote (SL2), Particles with a diameter of 2 μm (L3030), Fluorescein-5(6)-iso-thiocyanate (F3651) and 4-Carboxy-TEMPO (382000) were purchased from Sigma-Aldrich (Saint Louis, MO, USA). Hyaluronic acid (5 kDa) was purchased from SK-bioland (Seoul, South Korea) 5(6)-SFX (6-(Fluorescein-5-(and-6)-Carboxamido) Hexanoic Acid, Succinimidyl Ester) (F2181) was purchased from Thermo Fisher (Waltham, MA, USA). 2% PFPE-PEG-PFPE tri-block copolymer surfactant (E2K0660) from RAN Biotechnologies, Inc. (Beverly, MA, USA) were used.

### Sample preparation

Protamine (10 mg/mL), εPL (10 mg/mL) and HA (10 mg/mL) were dissolved in 0.1 M sodium acetate buffer (pH 5.0). By varying the concentrations of NaCl, the LLPS of the HA-Protamine and HA-εPL was quantified by turbidity at room temperature after mixing for 1 min. Turbidity measurements were carried out at 600 nm using UV–Vis spectrometer (Optizen, Daejeon, South Korea), where absorbance interference by HA, protamine, and εPL was negligible. Each experiment was performed three times. The formation of droplets was confirmed with optical microscopy (BX63, Olympus, Tokyo, Japan). If not stated otherwise, all samples for HA-Protamine and HA-εPL coacervation were prepared under charge-balanced optimal phase separation conditions of 7:3 and 8:2 ratios, respectively (Supplementary Fig. 10).

### Temperature-dependent phase behavior of coacervates

Absorbance of HA-protamine coacervation was measured with a temperature controllable circular dichroism (CD) spectrometer (J-850, JASCO, Tokyo, Japan) at 600 nm. The temperature was changed from 4 °C to 70 °C at a rate of 2 °C/min. The experiment was conducted with heating and cooling cycles.

### Quantitative amino acid analysis

To determine the protamine and hyaluronic acid concentrations in the droplets, a quantitative amino acid analysis was performed with a SYKAM System S4300 Amino Acid Analyzer (SYKAM, Gewerbering, Germany), according to the previously reported method[64]. Specifically, protamine and hyaluronic acid were mixed at a ratio of 3:7 and then centrifuged at 920 × $g$ for 1 h to obtain the dilute phase and sedimented droplet phase. Each phase was then evaporated, and the dried pellets dissolved in 6 N HCl with 5% water-saturated phenol, and transferred to glass vials. The glass vials with sample were purged with argon and then sealed with flame. The acid hydrolysis reaction was carried out for 24 h in a 110 °C heating block. Afterward, the samples were evaporated and sequentially washed with DW and methanol. The samples were reconstituted in SYKAM sampled dilution buffer. To generate a standard curve using known concentrations of each polyelectrolyte, the same procedures were performed with four different concentrations of protamine solution, hyaluronic solution, and blank water. The peak areas of arginine and glucosamine for protamine and hyaluronic acid minus the mean corresponding peak area of the hydrolyzed water blank were calculated. Linear regression was applied to this adjusted peak area, correlated with the corresponding concentration of the four standard concentrations injected. Slopes were based on four-point calibration. These slopes were used to calculate the concentrations of each polyelectrolyte in the droplets and the dilute phase.

### Fluorescence recovery after photobleaching (FRAP)

FRAP experiments were conducted using a Leica TCS SP5 microscope (Wetzlar, Germany). HA-protamine droplets containing 2% FITC-protamine and HA-εPL droplets containing 2% TxRed-εPL were prepared for this measurement. Samples of 512 × 512 pixels were imaged with a 25× water-immersion objective. Full droplets were bleached with a 488 nm laser at 100% power. Recovery was monitored every 0.523 s for 400 frames. Images were processed by ImageJ. All imaging was performed with the same acquisition settings for optical zoom, scan speed, gain, resolution, offset, magnification, and laser intensity.

### Pulsed-field gradient NMR

Pulsed-field gradient NMR were carried out using a Bruker (Billerica, MA, USA) ULTRASHIELD 300 SWB PLUS spectrometer and a DIFF50 diffusion probe at a proton NMR frequency of 300.15 MHz. The experiments were operated by tunning the diffusion probe to $^1$H nuclei[i].H diffusion was measured on dense phase and dilute phase at room temperature. The diffusion coefficients of interstitial water are determined using a pulse sequence of stimulated echoes with bipolar pulses. The echo attenuation Ψ was fit to the Stejskal-Tanner equation, $\psi = \exp(-(\gamma g \delta)^2 D(\Delta - \frac{\delta}{3}))$, where $\gamma$ [$s^{-1} G^{-1}$] is the gyromagnetic ratio of protons, $g$ [G/cm] is the magnitude of the applied field gradient, $\delta = 1$ ms denotes the duration of the effective gradient pulse, $\Delta = 20$ ms is the total diffusion time, and $D$ [$m^2 s^{-1}$] is the diffusion coefficient. A total of 16 echoes were acquired using a range of gradient amplitudes for each measurement and measured attenuations were fitted with single-exponential decays. The self-diffusion coefficient of pure water (DW) is in good agreement with the literature value of $2.40 \times 10^{-9}$ m²/s[65].

### Spin labeling for EPR and ODNP

Protamine and εPL were labeled by 3-(ethyliminomethylideneamino)-N,N-dimethylpropan-1-amine-N-hydroxysuccinimide (EDC-NHS) chemistry with a 10-fold molar excess of the 4-carboxy-TEMPO spin label. Excess label was removed with extensive dialysis (3500 Da MWCO) for 24 h. Since 4-carboxy-TEMPO was labeled on the amine group, protamine has a spin only at the N-terminal, and spins are located throughout the εPL chain. Non-labeled protamine was used to achieve spin dilution. Spin-labeled (SL)-protamine was mixed with non-labeled protamine at a ratio of 1:3 to achieve 25% spin labeling. Then, the samples were prepared by adding hyaluronic acid.

### Electron paramagnetic resonance

Electron paramagnetic resonance (EPR) experiments were performed using an X-band (~0.35 Tesla; ~9.85 GHz) Bruker EMXPlus spectrometer equipped with a Bruker ER4119HS-W1 high sensitivity microwave cavity (Bruker) at room temperature. Samples of 4.0 μL was loaded into a quartz capillary of 0.6 mm ID × 0.84 mm OD (CV6084; Vitrocom, Mountain Lakes, NJ, USA), which were then sealed at both ends with Critoseal, and placed into 4 mm diameter open-ended quartz EPR tubes. EPR spectra were acquired using a microwave power of 4 mW, a modulation frequency of 100 kHz, a modulation amplitude of 1G, and a sweep width of 150G. EPR spectra were analyzed to determine the isotropic rotational correlation time of the spin label attached to the polyelectrolytes using the EasySpin software package[66] operating in MATLAB (Mathworks, Natick, MA, USA). The spectra were analyzed using the 'chili' function by first setting the g values and hyperfine coupling tensor values according to known literature values with minor adjustments to achieve an exact match of the positions of the three features of the nitroxide spectrum on the magnetic field axis. Next, the spectra were fit using the 'esfit' graphical user interface by varying Gaussian and Lorentzian line broadening parameters and the 'logtcorr' parameter that is the $\log_{10}$ of the isotropic rotational correlation time ($\tau_{corr}$). To first set the line broadening parameters the spectrum that was clearly indicative of the least restricted spin label motion (i.e minimal broadening and minimal decrease in amplitude of the high-field feature of the nitroxide spectrum) was fit by varying the

line broadening parameters alone. Since the simple line broadening parameters do not affect the spectrum in the same fashion as changes to $\tau_{corr}$ this results in a less than ideal fit but a good match to the intrinsic line broadening of the spectra due to experimental factors. This spectrum along with the others were then fit in series by varying only the 'logtcorr'; with the $g$ values, hyperfine coupling tensor values, and line broadening parameters held constant. With this, the trend in $\tau_{corr}$ between the various samples is captured. All EasySpin operations were executed within the cw-EPR application available for free at the MathWorks File Exchange.

## Overhauser dynamic nuclear polarization

For Overhauser dynamic nuclear polarization (ODNP) experiments, 3.5 µL of dilute and dense-phase spin-labeled protamine samples were loaded into quartz capillaries of 0.6 mm ID × 0.84 mm OD (CV6084; Vitrocom), and both ends of the tubes were sealed with Critoseal. The capillary tube was mounted in a home-built NMR probe consisting of a tuning/matching circuit and a handmade U-shaped NMR coil. The coil was positioned in the center of a Bruker ER4119HS-W1 EPR resonator. A Bruker EMXplus EPR spectrometer was used to tune and match the EPR resonator and collect spectra as described above, but with the sample loaded into the NMR coil to account for the shift in tuning frequency and the resulting shift in magnetic field position of the EPR resonances. For the ODNP experiments, the magnetic field was set to the field value at the center of this EPR spectrum (-0.3484 T), and high-power microwaves for DNP were supplied by a custom microwave source at the tuning frequency of the resonator (~9.78 GHz). Dry nitrogen was passed over the sample to maintain a sample temperature of ~295 K. NMR spectra were collected using a Bruker Avance III NMR console (Bruker). ODNP data were analyzed using an open-source python-based software package called DNPLab designed for processing and analyzing NMR data. In particular, data were processed and fit using the 'hydrationGUI' module. This module applies the analysis procedures pioneered in the Han lab that have been previously described and applied in several similar studies[53,67–69].

## Microrheology

Microrheology was performed by tracking the motions of probe particles embedded in coacervate phase[57,69]. The fluorescent probe particles with a diameter of 2 µm were imaged with a confocal microscope at 561 nm excitation every 20 ms for low salt conditions and every 1 s for high salt conditions at room temperature. Mean-squared displacements (MSD) of the particles in dense phase was obtained using MATLAB software. The averaged MSD vs. lag time plot fit to MSD (τ) ~ $4D\tau^\alpha$, where α is the diffusive exponent, to estimate the diffusion coefficient $D_{probe}$. The viscosity η was calculated using the Stokes-Einstein equation, $D_{probe} = k_BT/6\pi\eta r$, where $k_B$ is the Boltzmann constant, $T = 295.15$ K is temperature, and $r = 1$ µm is the probe radius.

## Interfacial tension measurement by droplet coalescence

The interfacial tension of the HA-protamine coacervates was measured by observing coalescence events of two droplets over time. A solution containing droplets of FITC-tagged protamine-HA was added into a coverslip-sandwiched fluid chamber with a flat oil/water interface to minimize friction from the surface during droplet coalescence. Coalescence events were recorded with confocal microscopy with specific time intervals at 488 nm excitation. The decay time (τ) was determined with a ratio of the difference and sum of the length ($L$) and width ($W$) of a droplet during the relaxation, $A = \frac{L-W}{L+W}$. The interfacial tension γ of the droplet was determined from the time scale of the progress of the relaxation via, $\tau \cong \frac{19}{20}\frac{\eta R}{\sigma}$ where η is viscosity of the dense phase determined by microrheology, and R is the droplet radius after the coalescence.

## Molecular dynamics simulations method

The peptides were simulated using version 4.5.3 of the GROMACS molecular dynamics package[70]. The leapfrog integrator[71]. was used to integrate the equations of motion with a time-step of 2 fs. The oxygen-hydrogen bonds in water were constrained using the SETTLE algorithm[72], and all other bonds to hydrogens were constrained using the LINCS algorithm[73]. The modified tip4p/2005s water model was used, and the peptides were simulated using the Amber ff03ws force field[46]. We use the parametrized NaCl force field by Benavides et al., to prevent unphysical salt crystallization[74]. Short-range van der Waals and Coulombic interactions were truncated using a cut-off of 1.0 nm, and long-range electrostatics were calculated using the particle-mesh Ewald (PME) algorithm[75]. The simulations were performed in the iso-baric isothermal ensemble (NPT); the temperature and pressure were maintained at $T = 300$ K and $P = 1$ bar, using the stochastic velocity-rescale thermostat and Parrinello-Rahman Barostat[76]. In INDUS, the GROMACS package is modified to bias the coarse-grained water number, Nv, in hydration volume using the Indirect Umbrella Sampling (INDUS) method[44,77]. The number of water molecules in the hydration shell of each residue can be conformation dependent, and it can also change by the definition of the hydration volume itself. Here, we chose the size of the spherical probe volume of every heavy atom in such a way that we have at least two-three layers of water molecules in the hydration shell (the first coordination shell). We note that a careful selection of the probe volume is important; indeed, if the spherical probe volumes are very large, the resulting hydration volume would be less relevant to the surface of the target residue and the dewetting free energy can be largely affected by the errors due to the hydration water fluctuation. The Gaussian coarse-graining function used in INDUS is parameterized with a standard deviation of $\sigma = 0.01$ nm and a truncation length $r_c = 0.02$ nm. The water dynamics is governed by the Hamiltonian: $H = H_0 + 1/2\ \kappa\ (N_v - N^*)^2$, where $H_0$ is the unbiased Hamiltonian, and the second term represents the harmonic biasing potential with strength $\kappa = 10\ k_BT$. The biased ensemble averages $\langle N_v \rangle$ was obtained by sampling directly from biased ensembles with different N* values. All biased simulations were run for a total of 5 ns; the first 1 ns were discarded for equilibration phase. The overlapping successive windows of the $N_v$ (the instantaneous water molecules in the hydration volume) that were obtained from performing biased simulations at different N* (i.e. "for N* in {−9..3..54}") (Supplementary Fig. 11), enable us to estimate the dehydration free energy of the target residue. The dehydration free energy is determined by taking the minus logarithm of the unbiased probability distribution of observing N water molecules in the hydration volume of the target residue i.e., $\beta F(N) = -\ln P_v(N)$. The $P_v(N)$ is calculated using the Unbinned Weighted Histogram Analysis Method (UWHAM).

## Reporting summary

Further information on research design is available in the Nature Portfolio Reporting Summary linked to this article.

# Data availability

The datasets generated in this study are available in the Figshare repository under https://doi.org/10.6084/m9.figshare.21509343.v3. UniProt Knowledgebase (UniProtKB) (Proteome ID UP000005640, FUS, P35637, TAF15, Q92804, EWS, Q01844, hnRNPA1, P09651, G3BP1, Q13283, FMRP1, Q06787).

# Code availability

All ODNP experimental data were processed using DNPLab, an open-source Phyton package (Timothy Keller, Thomas Casey, Yanxian Lin, Thorsten Maly, & jmfranck. (2021). DNPLab/DNPLab: (Version 2DODNPPaper). Zenodo. https://doi.org/10.5281/zenodo.4670553). The INDUS code is open source and available on GitHub (https://github.com/patellab511/indus).

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

## Acknowledgements

The work of D.S.H. was supported by the National Research Foundation of Korea(NRF) grant funded by the Korea government (MSIT) (Nos. 2019M3C1B7025093 and 2022R1A2C2007874). The authors acknowledge support from the Center for Scientific Computing at the California Nanosystems Institute (CNSI, NSF grant CNS-1725797) for the availability of high-performance computing resources and support. Support for the ODNP facility was provided by the Deutsche Forschungsgemeinschaft (DFG, German Research Foundation) under Germany's Excellence Strategy—EXC-2033—Project No. 390677874.This work used the Extreme Science and Engineering Discovery Environment, which is supported by the National Science Foundation grant number ACI-1548562 (MCA05S027). J.-E.S. acknowledges support from the NSF (MCB-1716956). J.-E.S. and S.H. acknowledge support from the NIH (R01-GM118560-01A).

## Author contributions

Y.H. performed phase separation experiments and data analysis; S.N. performed computational simulations; T.C. performed EPR and ODNP experiments; Y.H. and D.S.H designed the project; J.-E.S., S.H., and D.S.H. supervised; Y.H., S.N., T.C., J.-E.S., S.H., and D.S.H. wrote and edited the manuscript.

## Competing interests

The authors declare no competing interest.
