## [Peer Review File · Nature Communications]

REVIEWER COMMENTS

Reviewer #1 (Remarks to the Author):

This manuscript presents a systematic study of the liquid liquid phase separation (LLPS) characteristics of two model systems protamine/ hyaluronic acid (HA), which is arginine rich and poly lysine (ϵ PL)/HA, which consists of only lysines. The purpose of the study is to illustrate the differences between arg and lys which stem from the ability of arg to generate π - π and π -cation interaction whereas their electrostatics are similar. These hydrophobic type of interactions are expected to increase the LLPS propensity of arg-rich proteins. While these idea are known in the literature this work demonstrates the difference between lys only and arg rich model systems experimentally, highlighting the differences arising from the hydrophobic character of the latter.

The main difference between the two systems is that protamine/HA undergoes LLPS at both low and high salt concentration while ϵ PL/HA does not. This is attributed to the hydrophobic interactions which become effective once the electrostatic interactions are screened at high salt.

The author used an impressive large number of methods to characterize and highlight the hydrophobic interaction or their consequences, which is quite unique. This is a solid work, the manuscript is well organized and in general clear, it is important and of interest, though not very imaginative and results are of a confirmatory nature.

Below I list a few concerns.

1. Why did the author choose protamine for the comparative study and not poly-arginine ? While ϵ PL consists of only lysines, protamine has other amino acids, some of them hydrophobic (V and M) and P which is special - maybe they contribute to the differences observed and to the hydrophobicity of the protamine ? This is not discussed at all.
2. Why was the relative amount of the HA different in the two systems.
3. There is no information on the source of chemicals used .
4. Was FRAP attempted also on the protamine/HA high salt ?
5. The MD simulations are carried out on single peptide single peptides in the presence of salt and conclusions regarding the conformation of the peptide are derived and used to account for LLPS behavior. Why would a more closed (compact) conformation at high salt enhance LLPS where transient interaction between peptides are needed? Did the author attempt to carry out MD simulations on several chains, in the presence of HA (small fragments) ? These would be more relevant.
6. The EPR figure is incomprehensible – the caption does not mention what are the two overlaid spectra. Also, there two spectra for high salt case are overlaid with different scales. The rigid limit spectrum contains clear contributions from a highly mobile component, the lines of which do not overlap with those of the fast limit superimposed spectrum.
7. In line 320 the authors write about the rigid limit EPR spectrum “These results further validate that enhanced hydrophobic interactions from arginine impart material properties to arginine-rich phases of LLPS”. The slow rotational diffusion can arise from the high local viscosity due to the high salt concentration and does not necessarily point to hydrophobic interactions.
8. The authors claims that the hydrophobicity of arg tunes the reentrant LLPS behavior. I think that a better word would be “lead”, or “responsible”. The use of tune would have been appropriate if the effect of the number of arg residue would have been tested.
9. I find the comparison with the stress granules and the associated conclusions to be too speculative.

Reviewer #2 (Remarks to the Author):

The manuscript "Hydrophobicity of arginine tunes reentrant liquid-liquid phase separation behaviors of arginine-rich proteins" by Hong et al. investigates the role of arginine hydrophobicity in dictating the salt-mediated reentrant liquid-liquid phase separation (LLPS) of proteins. The authors combine experiments and atomistic molecular dynamics simulations to investigate the complex coacervation of the positively charged arginine-rich versus lysine-rich proteins with the negatively charged hyaluronic acid, in a low and high salt regime. I enjoyed reading this paper, and I would be supportive of its publication in Nature Communications. However, I have a few suggestions and comments that I would like the authors to address.

1. Many of the results presented in this work are expected. For instance, the unequal contribution of arginine vs lysine in phase separation at low and high salt, the higher hydrophobic nature of arg vs lys, the importance of pi-pi and cation-pi interactions that arginine can form and their strengthening with increasing salt, the impact of arginine and lysine in the viscoelastic properties of condensates, the divergent solvation free energy of arginine and lysine, etc. have all been well established, which have been demonstrated by works of Pappu, Mittag, Formann-Kay, Elbaum-Garfinkle, Knowles and Banjeree and their collaborators. Nonetheless, the results of this paper are valuable as they demonstrate that salt-mediated reentrant phase behavior of proteins also occurs during complex coacervation. Salt-driven reentrant behaviour had been previously reported and characterized by Krainer et al. for pure protein systems (i.e. where phase separation is driven by homotypic rather than heterotypic interactions, as in the present work). Consistently with the work by Krainer et al, the current paper reveals that salt-mediated reentrant complex coacervation of protein mixtures can be explained at the molecular level in an equivalent manner to what was proposed by Krainer et al. for pure proteins. That is, the low salt condensates are stabilized by a combination of electrostatics, cation-pi, pi-pi, and other hydrophobic interactions, while the high-salt reentrant condensates are fully stabilized by non-ionic interactions.

2. Now, what I think is novel in this paper and I was excited to read about is the connection between the temperature-dependent and the salt-dependent reentrant phase behaviour of arginine-rich complex coacervates. I think this observation is not given the prominence it should, and it's a missed opportunity. In particular, the discussion on why arginine-rich complex coacervates exhibit both UCST and LCST at low salt but only UCST at high salt, while lysine-rich coacervates only present LCST at low salt should be expanded. The authors already hypothesize that the LCST systems can be explained by a dominant entropic contribution due to water release upon phase separation, while the UCST regime might be explained by a dominant enthalpic contribution. While the hypotheses make sense, the reason why a system exhibits UCST vs LCST or both are not fully understood. Since this is perhaps the most interesting/novel point of the paper, I would encourage the authors to think about this result in more detail and provide further insight on the molecular mechanisms behind it. Given that they observe different behaviours of lysine vs arginine rich systems, can the contributions of hydrophobic and electrostatic forces and their modulation with salt be linked to their observations? Perhaps the authors could exploit their expertise in molecular simulations to test their hypothesis.

3. The authors use molecular simulations to estimate the context-dependent hydrophobicity of various arginine and lysine residues within protamine vs polyK. However, I am not very convinced about the results presented in this section. Converging the calculations of dehydration free energies of well-folded proteins is already a very challenging task, and here the authors deal with those calculations for IDPs. It is necessary therefore for the

authors to discuss the difficulties in these calculation, show how their estimates of number of water molecules fluctuate over time, and demonstrate how they managed to obtain reliable values for the dehydration free energies, which depend on the shape and size of the cavities formed near the protein and are expected to be very diverse in IDPs and change with the frequent fluctuations in protein configurations. This issue is further amplified because the results are given without any discussion of error estimations and error bars, and discussions on ensemble deviations are also missing. The main result of these calculations, i.e., the difference of 0.6kT for the energy cost of removing a water molecule from the shell of Arg vs Lys, is quite small and potentially similar to the size of the errors in the calculation. It is also not clear why it is claimed that their calculations reveal that Arg in protamine is more hydrophobic than Lys in polyK at the residue level, when all the curves in Fig S2 for both Arg and Lys look very similar, and indeed differences among the various Arg curves seem comparable to the difference between Lys and Arg. The INDUS method is not standard, and the paper is aimed at an interdisciplinary audience; thus, I would expect a more conceptual and thorough explanation of why they used this approach, how it works, and what it measures.

4. The authors discuss that the densities of protamine and hyaluronic acid measured experimentally are greater at high salt than at low salt. However, given that the density is heavily modulated by temperature too, the authors should explain the context dependency of these results. By changing only the salt concentration but keeping T fixed, the authors are moving from one arbitrary point in salt and temperature in the phase diagram of the system to a second arbitrary point on the phase diagram. Thus, there is no reason whatsoever to expect the system to exhibit the same densities. To gain a better understanding on the relationship of density variations on the two regimes, the authors would need to measure densities on multiple combinations of temperatures and salt concentrations. The same can be said for the results on the physical properties of the dense coacervate phases.

5. When using molecular simulations again to estimate the radii of gyration and end-to-end distance, I wonder which parameters for ions did the authors used. The parameters for ions in force fields are not necessarily well-fitted to give reasonable ion solubilities at room temperature. This is typically not an issue when performing molecular simulations at physiological salt concentrations, which are the standard. However, at the high salt concentrations used here to mimic the reentrant regime, some ions parameters can yield the formation of unphysical crystals. Thus, estimating the ion solubility of the model is crucial.

Reviewer #3 (Remarks to the Author):

The manuscript reports results from experiments in which the physicochemical properties of complex coacervates formed by an arginine-rich or an lysine-rich peptide with an oppositely charged factor. The authors state that a major insight from this work is the importance of difference in hydrophobicity between arginine and lysine for complex coacervation-based phase separation.

Overall the experiments are well done and partially supported by molecular dynamics simulations. However, the study has two major limitations.

First, the conclusion regarding the important role of hydrophobicity is most strongly based on the differences in phase separation behaviour of the arginine- and lysine-rich peptide at very high salt. This however cannot be translated back to physiologically relevant salt

conditions where it is likely that a combination of different factors (maybe hydrophobicity, but likely also others as reported already in the literature) are important. Thus, the physiological relevance of the current data is questionable.

Second, several studies already have shown the different roles of arginine and lysine for peptide/protein phase separation. This also includes for example reference #25, in which it was shown that the viscoelasticity of arginine-based condensates is greater than that of lysine-based condensates. Thus the novelty/broad interest of the results from the current study is limited.

The manuscript is more suited for a journal with a focus on physical chemistry (i.e. J Phys Chem).

Response Letter to Reviewers

We would like to thank the reviewers for their time and supportive comments. We have revised the manuscript in accordance with their comments and concerns, as detailed below. The responses to all comments have been prepared and given below. The original reviewers' comments are provided in *black color*, whereas our answers are given in *blue*. The appropriate changes made in the revised manuscript are **highlighted**. We have also taken the opportunity to make a few additional changes to clarify some points and to enhance the quality of the manuscript; these are also **highlighted**. The highlighted sections of the manuscript are reproduced in our response below in *red* font.

Reviewer #1 (Remarks to the Author):

This manuscript presents a systematic study of the liquid liquid phase separation (LLPS) characteristics of two model systems protamine/ hyaluronic acid (HA), which is arginine rich and poly lysine (ϵ PL)/HA, which consists of only lysines. The purpose of the study is to illustrate the differences between arg and lys which stem from the ability of arg to generate π - π and π -cation interaction whereas their electrostatics are similar. These hydrophobic type of interactions are expected to increase the LLPS propensity of arg-rich proteins. While these idea are known in the literature this work demonstrates the difference between lys only and arg rich model systems experimentally, highlighting the differences arising from the hydrophobic character of the latter.

The main difference between the two systems is that protamine/HA undergoes LLPS at both low and high salt concentration while ϵ PL/HA does not. This is attributed to the hydrophobic interactions which become effective once the electrostatic interactions are screened at high salt.

The author used an impressive large number of methods to characterize and highlight the hydrophobic interaction or their consequences, which is quite unique. This is a solid work, the manuscript is well organized and in general clear, it is important and of interest, though not very imaginative and results are of a confirmatory nature.

Below I list a few concerns.

RIQ1. *Why did the author choose protamine for the comparative study and not poly-arginine? While ϵ PL consists of only lysines, protamine has other amino acids, some of them hydrophobic (V and M) and P which is special - maybe they contribute to the differences observed and to the hydrophobicity of the protamine? This is not discussed at all.*

RIA1. Thank you for your comments. As you have pointed out, protamine has other amino acids. Nevertheless, we chose protamine as an arginine-rich model protein. This is because protamine can be utilized as a biomedical material. Protamine is a clinically relevant, biocompatible, and injectable biopolymer. It has been used as a drug to neutralize the anticoagulant effects of heparin and to formulate slow-release insulin. Furthermore, it has been used as a transfection enhancement agent for RNA delivery. Thus, our study of protamine could give more direct insight into the application of the protamine phase-separating system.

As a response to the reviewer's question, we have now performed MD simulations of poly-arginine. Our results demonstrate that protamine has hydrophobic (Supplementary Fig. 3) and conformational properties (Supplementary Fig. 6 and 7) similar to those of poly-arginine. As in the case of protamine (Fig. 6e, f and Supplementary Fig. 5b, d), poly-arginine features a global minimum corresponding to extended conformations in the absence of salt, and the population is skewed towards the collapse conformations at high salt conditions. We now discuss this comparison in greater detail.

Fig. 6e, f & Supplementary Fig. 5b, d

Supplementary Figure 7 has been added:

Supplementary Figure 7. Conformational properties of poly-arginine. **a** and **b** represent the probability distribution of the radius of gyration (R_g) and the end-to-end (d_{ee}) distance of

poly-Arginine without and with salt, respectively. **c** and **d** show the intramolecular contact map of poly-Arginine in presence of salt and without salt, respectively.

Added to Results (Page 17, Line 384):

...This conformational property of protamine is similar to that of poly-arginine (Supplementary Fig. 7)

R1Q2. *Why was the relative amount of the HA different in the two systems.*

R1A2. Our goal was to compare HA-Protamine and HA- ϵ PL systems under charge-balanced optimal phase-separation conditions. Thus, we first check optimal HA: protein (Protamine or ϵ PL) mixing ratio by measuring turbidity using UV/Vis spectrometer. According to turbidity, HA-Protamine and HA- ϵ PL coacervation were prepared at a ratio of 7:3 and 8:2, respectively. Inevitably, the relative amount of HA is different in the two systems. As a response to the review, we add the following Supplementary Figure 9 that we refer to in the manuscript.

Supplementary Figure 9 has been added:

Supplementary Figure 9. Turbidity measurement of HA-Protamine and HA- ϵ PL coacervation depending on the mixing ratio. Total concentration is 10 mg/mL. N=3, Error bar=S.D.

Added to Methods (Page 25, Line 564):

... If not stated otherwise, all samples for HA-Protamine and HA-εPL coacervation were prepared under charge-balanced optimal phase separation conditions of 7:3 and 8:2 ratios, respectively (Supplementary Fig. 9).

R1Q3. *There is no information on the source of chemicals used.*

R1A3. Thank you for pointing this out. We have now included information on the source of chemicals.

Added to Methods (Page 25, Line 546):

Materials. Protamine sulfate salt from salmon (5.1kDa, p4020, Lot# SLBW4512, *Oncorhynchus keta*), Poly(ethylene glycol) (10kDa), Sigmacote (SL2), Particles with a diameter of 2μm (L3030), Fluorescein 5(6)-isothiocyanate (F3651) and 4-Carboxy-TEMPO (382000) were purchased from Sigma-Aldrich (Saint Louis, MO, USA). Hyaluronic acid (5kDa) was purchased from SK-bioland (Seoul, Korea) 5(6)-SFX (6-(Fluorescein-5-(and-6)-Carboxamido) Hexanoic Acid, Succinimidyl Ester) (F2181) was purchased from Thermo Fisher (San Jose, USA). 2% PFPE-PEG-PFPE tri-block copolymer surfactant (E2K0660) from RAN Biotechnologies, Inc. (Beverly, MA, USA) were used.

R1Q4. *Was FRAP attempted also on the protamine/HA high salt?*

Thank you for this suggestion. We agree with your suggestion and have incorporated your suggestion in Fig. 6a. Consistent with the EPR results, the exchange of protamine molecules between the dilute and dense phases was greatly reduced for high salt induced HA-Protamine coacervation.

Figure. 6a has now been corrected as follows:

Fig. 6a FRAP curve of entire droplet of HA-Protamine (n=4) and HA-εPL (n=3) complex coacervation without additional salts and with 4 M NaCl. Error bar = s.d. (Top). Representative images (Bottom).

Added to Results (Page 15, Line 323):

…However, at the high salt concentration, the exchange of protamine molecules between the dilute and dense phases was greatly reduced (Fig. 6a, Bottom).

R1Q5. *The MD simulations are carried out on single peptide single peptides in the presence of salt and conclusions regarding the conformation of the peptide are derived and used to account for LLPS behavior. Why would a more closed (compact) conformation at high salt enhance LLPS where transient interaction between peptides are needed? Did the author attempt to carry out MD simulations on several chains, in the presence of HA (small fragments) ? These would be more relevant.*

R1A5. We thank the reviewer for raising this question.

The conformational property of the peptide is strongly linked to its collective behavior (in a large ensemble) such as aggregation and LLPS. We showed in earlier work on KE(K=Lysine, E=Glutamic acid) peptides that peptides with sequences that favor compact conformations can in fact undergo LLPS more readily than sequences that favor extended conformations (see reference: *J. Phys. Chem. Lett.* 2019, 10, 8, 1644–1652 Complete Phase Diagram for Liquid–Liquid Phase Separation of Intrinsically Disordered Proteins). The factors that drive compaction of the monomeric state are also those that drive the LLPS process. Thus, the intramolecular driving force (that tunes the conformational behavior of a monomer), can, when other peptides are present, be readily transformed to intermolecular interactions that drive LLPS. We have clarified this issue in the revised version of the manuscript as follows.

Added to Results (Page 16, Line 352):

... The conformational features of the peptide, governed by intramolecular interactions, can influence collective behaviors of peptides, such as their aggregation and LLPS properties. For example, strong intramolecular interactions that lead a single peptide to adopt more collapsed globular states, can, in the presence of other peptides, can also facilitate intermolecular interactions that drive LLPS (reference).

Reference:

Mccarty, J., Delaney, K. T., Danielsen, S. P. O., Fredrickson, G. H. & Shea, J.-E.. Complete Phase Diagram for Liquid–Liquid Phase Separation of Intrinsically Disordered Proteins. *J. Phys. Chem. Lett.* **10**, 1644 (2019).

R1Q6. *The EPR figure is incomprehensible – the caption does not mention what are the two overlaid spectra. Also, there two spectra for high salt case are overlaid with different scales. The rigid limit spectrum contains clear contributions from a highly mobile component, the lines of which do not overlap with those of the fast limit superimposed spectrum.*

R1A6. We agree that the presentation of the EPR data was unclear and confusing. We have corrected Fig. 6b to include the EPR spectra of spin-labeled proteins in the dense phase of low salt HA- ϵ PL, low salt and high salt HA-Protamine, and display them on the same scale. We also added Supplementary Figure 5, which contains all EPR spectra plotted on the same scale.

Figure. 6b has been corrected:

Fig. 6 **b** EPR spectra of spin-labeled ϵ PL and protamine in the dense phase of HA- ϵ PL complex coacervation (*Top*), HA-Protamine complex coacervation (*Middle*) without additional salts, and HA-Protamine complex coacervation under high salt condition (*Bottom*).

R1Q7. *In line 320 the authors write about the rigid limit EPR spectrum “These results further validate that enhanced hydrophobic interactions from arginine impart material properties to arginine-rich phases of LLPS”. The slow rotational diffusion can arise from the high local viscosity due to the high salt concentration and does not necessarily point to hydrophobic interactions.*

R1A7. The reviewer makes a good point. However, we did check that the spin-labeled protamine alone in the same high salt solution, but without HA, does not display a rigid limit EPR spectrum (Supplementary Fig. 5). We now include this data in Supplementary Fig. 5 in a row that we refer to as “Protein in buffer” as we show below.

Indeed, the rigid limit EPR spectral features only appear for HA-Protamine under high salt coacervation conditions, and hence we can attribute this to the coacervate internal materials property.

Supplementary Figure 5 has been added:

Supplementary Figure 5. EPR spectra of SL-εPL and SL-protamine in dense phase, dilute phase and low/high salt buffer (without coacervation).

R1Q8. *The authors claim that the hydrophobicity of arg tunes the reentrant LLPS behavior. I think that a better word would be “lead”, or “responsible”. The use of tune would have been appropriate if the effect of the number of arg residue would have been tested.*

R1A8. Thanks for the suggestion. We modified the title to “Hydrophobicity of arginine leads to reentrant liquid-liquid phase separation behaviors of arginine-rich proteins”.

R1Q9. *I find the comparison with the stress granules and the associated conclusions to be too speculative.*

R1A9. We agree that we should make it clear that we are speculating, so we have added Supplementary Figure 8 showing that several phase-separating stress granules proteins are rich in arginine.

Supplementary Figure 8 has been added:

Supplementary Figure 8. RGG/RG motif-containing phase-separating proteins in stress granules. **a** Arginine to lysine ratio of the full sequence (grey) and intrinsically disordered regions (IDRs) (pink) of the RGG/RG motif-containing SG proteins compared to the arginine to lysine ratio of the human proteome. **b** Arginine percentage of the full sequence (grey) and IDRs (pink) of the RGG/RG motif-containing SG proteins compared to the arginine percentage of the human proteome. Human proteome database is obtained in UniProt Knowledgebase (UniProtKB) (Proteome ID UP000005640). IDRs is determined based on the UniProtKB database. IDRs (amino acid sequence) of FUS (RNA-binding protein FUS): 1-286/375-424/444-526, TAF15 (TATA-binding protein-associated factor 2N): 1-237/324-356/373-592, EWS (Ewing sarcoma breakpoint region 1 protein): 123-360/448-525/547-656, hnRNPA1 (Heterogeneous nuclear ribonucleoprotein A1): 182-240/317-372, G3BP1 (Ras GTPase-activating protein-binding protein 1): 144-172/184-431/255-329/413-466, FMRP1 (Fragile X mental retardation protein 1): 325-349/443-632

We have also corrected the Discussion about the stress granules (Page 23, Line 526):

Last, but not least, arginine is also prevalent in RNA-binding proteins (RBPs), which are one of the main components of phase-separated stress granule (SGs) (Supplementary Figure 8)²⁰⁻²². Under physiological conditions, SGs have been understood to be involved in storing untranslated mRNAs, resulting in translational arrest⁵⁷. Based on previous observations, SG-related RBPs with disordered arginine-rich motifs undergo phase separation in vitro and in vivo^{10,11,58,59}. However, SGs induced and co-localized by arginine-rich peptides display reduced dynamic properties^{21,60} and sometimes, gelation or aggregation of SGs is known to be associated with neurodegenerative diseases⁵⁷. In agreement with this, our results show that the high salt HA-Protamine coacervates driven by hydrophobic interactions have solid-

like properties of elasticity and high immobility of protamine and water, along with an increase of protamine concentration within the coacervate phase. Additionally, at physiological low salt concentrations, arginine-rich protein coacervates are much denser and exhibit viscoelastic properties, but still contain dynamic protein chains and water comparable to lysine-rich protein coacervates. This appears to indicate that arginine-rich domains/proteins of SGs might function to retain and protect mRNA by generating tight binding and viscoelastic environment for mRNA. ~~Our study also shows that the dynamic property of the protein and water, and by extension the SG fluidity and function, are maintained.~~ We hence speculate that the activity and function of biomolecular constituents can be maintained in engineered arginine-rich membraneless organelles or biomolecular condensates.

Reviewer #2 (Remarks to the Author):

The manuscript “Hydrophobicity of arginine tunes reentrant liquid-liquid phase separation behaviors of arginine-rich proteins” by Hong et al. investigates the role of arginine hydrophobicity in dictating the salt-mediated reentrant liquid-liquid phase separation (LLPS) of proteins. The authors combine experiments and atomistic molecular dynamics simulations to investigate the complex coacervation of the positively charged arginine-rich versus lysine-rich proteins with the negatively charged hyaluronic acid, in a low and high salt regime. I enjoyed reading this paper, and I would be supportive of its publication in Nature Communications. However, I have a few suggestions and comments that I would like the authors to address.

R2Q1. Many of the results presented in this work are expected. For instance, the unequal contribution of arginine vs lysine in phase separation at low and high salt, the higher hydrophobic nature of arg vs lys, the importance of pi-pi and cation-pi interactions that arginine can form and their strengthening with increasing salt, the impact of arginine and lysine in the viscoelastic properties of condensates, the divergent solvation free energy of arginine and lysine, etc. have all been well established, which have been demonstrated by works of Pappu, Mittag, Formann-Kay, Elbaum-Garfinkle, Knowles and Banjeree and their collaborators. Nonetheless, the results of this paper are valuable as they demonstrate that salt-mediated reentrant phase behavior of proteins also occurs during complex coacervation. Salt-driven reentrant behaviour had been previously reported and characterized by Krainer et al. for pure protein systems (i.e. where phase separation is driven by homotypic rather than heterotypic interactions, as in the present work). Consistently with the work by Krainer et al, the current paper reveals that salt-mediated reentrant complex coacervation of protein mixtures can be explained at the molecular level in an equivalent manner to what was proposed by Krainer et al. for pure proteins. That is, the low salt condensates are stabilized by a combination of electrostatics, cation-pi, pi-pi, and other hydrophobic interactions, while the high-salt reentrant condensates are fully stabilized by non-ionic interactions.

R2A1. Thank you for your careful reading of the manuscript and your constructive remarks.

R2Q2. Now, what I think is novel in this paper and I was excited to read about is the connection between the temperature-dependent and the salt-dependent reentrant phase behaviour of

arginine-rich complex coacervates. I think this observation is not given the prominence it should, and it's a missed opportunity. In particular, the discussion on why arginine-rich complex coacervates exhibit both UCST and LCST at low salt but only UCST at high salt, while lysine-rich coacervates only present LCST at low salt should be expanded. The authors already hypothesize that the LCST systems can be explained by a dominant entropic contribution due to water release upon phase separation, while the UCST regime might be explained by a dominant enthalpic contribution. While the hypotheses make sense, the reason why a system exhibits UCST vs LCST or both are not fully understood. Since this is perhaps the most interesting/novel point of the paper, I would encourage the authors to think about this result in more detail and provide further insight on the molecular mechanisms behind it. Given that they observe different behaviours of lysine vs arginine rich systems, can the contributions of hydrophobic and electrostatic forces and their modulation with salt be linked to their observations? Perhaps the authors could exploit their expertise in molecular simulations to test their hypothesis.

R2A2. Thank you for bringing out attention to this point.

We describe our reasoning here that is also reflected in revision of our manuscript text. Electrostatic attractive interactions between oppositely charged polyelectrolytes accompanied by counterion release and hydration water release give rise to the entropic contribution to coacervation [1,2]. In this case, the tendency of LLPS increase with increasing temperature. The UCST behavior of HA-Protamine can be explained by the enthalpic contribution from hydrophobic self-interactions of arginine molecules through π - π stacking of guanidinium side chain at low temperature. This is because arginine π - π stacking is known to decrease with temperature (UCST behavior) [3].

Moreover, here we determine the distribution of the number of hydrogen bonds between the hydration water and the peptides. As shown in Supplementary Fig. 2, the number of hydrogen bonds between protamine and hydration water is larger than that of poly-Lys, in the absence of the salt. Thus, one can expect that the interplay between the entropic(dehydration) and enthalpic interactions that tunes the LCST and UCST behaviors at low salt, must be more pronounced in the case of protamine. In contrast, at high salt concentration, the probability distributions of the hydrogen bonds between the hydration water and the peptides are almost identical between protamine and poly-Lys, indicating that the entropic term does not play an important role on the LLPS behavior at high salt.

At low salt concentrations, HA-Protamine coacervation is driven by both hydrophobic and electrostatic interactions. Thus, the dual UCST and LCST behavior of low salt HA-Protamine

coacervation are attributed to the enthalpic contribution of the hydrophobic arginine π - π interaction and the entropic contribution of the electrostatic interaction accompanied by counterion release and the hydration water release, respectively.

In contrast, HA-Protamine coacervation is not driven by electrostatic interactions at high salt concentrations, where the entropic term (counterion release and dehydration) is insignificant. Thus, with the predominance of enthalpic contributions of hydrophobic arginine π - π interaction, high salt HA-Protamine coacervation exhibit only UCST behavior.

Reference:

[1]Chang, L.-W. *et al.* Sequence and entropy-based control of complex coacervates. *Nat. Commun.* **8**, 1273 (2017).

[2]Park, S.. Dehydration entropy drives liquid-liquid phase separation by molecular crowding. *Commun Chem* **3**, 83 (2020).

[3]Zydziak, N. *et al.* Unexpected aqueous UCST behavior of a cationic comb polymer with pentaarginine side chains. *Eur Polym J* **125**, 109528 (2020)

This interpretation has been added to Results (Page 9, Line 202):

...Commonly, electrostatic interactions are accompanied by counterion release and hydration water release upon coacervation, giving rise to entropic contributions that drive LCST behavior (ref1,2). Concurrently, the hydrophobic self-interaction of arginine molecules through π - π stacking of guanidinium side chain is at play that is known to decrease with temperature, giving rise to enthalpic contribution that drive UCST behavior (ref3). Thus, HA-Protamine coacervation at low salt concentrations, driven by the action of both hydrophobic and electrostatic interactions, display dual UCST and LCST behavior. In contrast, for HA-Protamine coacervation at high salt concentrations, it shows only UCST behavior because electrostatic interactions are screened out and hence entropic contribution to coacervation is insignificant. In the absence of salt, the number of hydrogen bonds between the hydration water and protamine is much greater than in the case of poly-Lys (Supplementary Fig. 2), implying that the release of hydration water upon coacervation of HA-Protamine can much more effectively contribute to entropy-driven LCST behavior at low salt concentration. In contrast, at high salt concentration, the number of hydrogen bonds between hydration water and the polymer is comparable between protamine and poly-Lys. Instead, at high salt concentration the dominant effect that drives HA-Protamine

UCST-LLPS is the enthalpic interaction among Arg residues. This is evidenced by the major increase of intramolecular contacts of Arg in the protamine chain upon increasing the salt concentration (Supplementary Fig. 6b, d).

Supplementary Figure 2. The probability distribution of the total hydrogen bonds between the hydration water and poly-Lysine, and the total hydrogen bonds between the hydration water and protamine without/with salt.

R2Q3. *The authors use molecular simulations to estimate the context-dependent hydrophobicity of various arginine and lysine residues within protamine vs polyK. However, I am not very convinced about the results presented in this section. Converging the calculations of dehydration free energies of well-folded proteins is already a very challenging task, and here the authors deal with those calculations for IDPs. It is necessary therefore for the authors to discuss the difficulties in these calculation, show how their estimates of number of water molecules fluctuate over time, and demonstrate how they managed to obtain reliable values for the dehydration free energies, which depend on the shape and size of the cavities formed near the protein and are expected to be very diverse in IDPs and change with the frequent fluctuations in protein configurations. This issue is further amplified because the results are given without any discussion of error estimations and error bars, and discussions on ensemble deviations are also missing. The main result of these calculations, i.e., the difference of $0.6kT$ for the energy cost of removing a water molecule from the shell of Arg vs Lys, is quite small*

and potentially similar to the size of the errors in the calculation. It is also not clear why it is claimed that their calculations reveal that Arg in protamine is more hydrophobic than Lys in polyK at the residue level, when all the curves in Fig S2 for both Arg and Lys look very similar, and indeed differences among the various Arg curves seem comparable to the difference between Lys and Arg. The INDUS method is not standard, and the paper is aimed at an interdisciplinary audience; thus, I would expect a more conceptual and thorough explanation of why they used this approach, how it works, and what it measures.

R2A3. We thank the reviewer for discussing this context dependent hydrophobicity of residues and how it becomes more complicated for IDPs. As the reviewer pointed out, the number of water molecules in the hydration shell of each residue can be conformation dependent, and it can also change by the definition of the hydration volume itself. However, as shown in Supplementary Figure 3, panel A, the Lys hydrophobicity in the homopolymer poly-Lys does not change across different sites and locations along the peptide. Here, we should also mention that the calculated free energies feature very small errors, i.e., less than 0.03kT (characterized by bootstrapping). Thus, if we re-perform the biased MD simulations of the investigated residues, we will get the same dewetting free energy within 0.03kT deviation; this is also reflected in the overlap of the Lys dewetting free energies along the poly-Lys chain (Supplementary Figure 3 panel a). In fact, we observe similar behavior if we investigate the Arg residue hydrophobicity in the homopolymer polyArg (Supplementary Figure 3 panel c).

In contrast, we see that the Arg dewetting free energy is a context dependent parameter in the protamine chain that is composed of different types of residues. This indicates that depending on the type of the adjacent residues, the hydrophobicity of Arg can change. However, surprisingly the average dewetting free energies (over the three investigated residues: 4, 16, 32) in the polyArg and protamine peptides are still identical (Supplementary Fig 3 panel d). Currently, in a separate project, we are investigating this behavior for a class of IDPs.

We agree with the reviewer about the effect of the shape of the probe volume. In this work, we chose the size of the spherical probe volume of every heavy atom in such a way that we have at least two-three layers of water molecules in the hydration shell (the first coordination shell). Here we are careful that if the spherical probe volumes are very large, the resulting hydration volume would be less relevant to the surface of the target residue and the dewetting free energy can be largely affected by the errors due to the hydration water fluctuation.

Although, we disagree with the reviewer that 0.6 kT dewetting free energy difference per-water molecule between Arg and Lys is small, it is indeed remarkable, because the total energy that we need to expel a water from the hydration shell of Arg and Lys in the extreme cases is about ~2.3kT and ~2.9kT. Considering that the average normalized water fluctuation (to N_0 , the total hydration water at equilibrium; for Arg₁₆ in protamine $N_0 = 44$) as shown in Supplementary Figure 10, is ~0.03 across different biased N^* , the effective dewetting free energy fluctuation per-water molecule is only ~0.06 kT, note that ~0.06 kT is 0.1 of the dewetting free energy difference between Arg and Lys.

We have carefully modified the manuscript accordingly to include all the points that the reviewer makes in this comment.

Added to Results (Page 12, Line 256):

...As illustrated in Supplementary Fig. 3b, the dewetting free energy of Arg in the protamine chain can vary along the sequence, depending on the type of the nearby residues. This nonidentical dewetting free energy behavior indicates the occurrence of context-dependent hydrophobicity of a particular residue in a peptide featuring a nonuniform chemical sequence. However, the Arg dewetting free energy remains unchanged in a homopeptide polyArg chain (Supplementary Fig. 3c) that mainly features expanded conformations with minimal structural complexity. Surprisingly, the average dewetting free energy of Arg in protamine and polyArg is identical (Supplementary Fig. 3d). This indicates that while the local hydrophobicity of a residue can be different along a complex peptide, its globally averaged dewetting free energy can still be similar to that of the residue in a homopeptide.

Added to Methods (Page 32, Line 722):

... The overlapping successive windows of the N_v (the instantaneous water molecules in the hydration volume) that were obtained from performing biased simulations at different N^* (i.e. “for N^* in { -9..3..54 }”) (Supplementary Fig. 10), enable us to estimate the dehydration free energy of the target residue. The dehydration free energy is determined by taking the minus logarithm of the unbiased probability distribution of observing N water molecules in the hydration volume of the target residue i.e., $\beta F(N) = -\ln P_v(N)$. The $P_v(N)$ is calculated using the Unbinned Weighted Histogram Analysis Method (UWHAM).

Supplementary Figure 3 has been corrected:

Supplementary Figure 3. **a** Dewetting free energies per water molecule that are independently calculated in the hydration volume of Lys₂, Lys₁₆ and Lys₃₃. **b**, The same quantity is characterized in the hydration volume of Arg₄, Arg₁₆ and Arg₃₃ of protamine chain. **c** The same quantity is characterized in the hydration volume of Arg₄, Arg₁₆ and Arg₃₃ of poly-Arg chain. **d** Average dewetting free energies in the polyLys, protamine, and polyArg.

Supplementary Figure 10 has been added:

Supplementary Figure 10. The time series of the hydration water of Arg₁₆ in protamine at different biased N^* (the biased hydration water). Note that N_v is the number of water molecules that we get after biasing N^* .

R2Q4. The authors discuss that the densities of protamine and hyaluronic acid measured experimentally are greater at high salt than at low salt. However, given that the density is

heavily modulated by temperature too, the authors should explain the context dependency of these results. By changing only the salt concentration but keeping T fixed, the authors are moving from one arbitrary point in salt and temperature in the phase diagram of the system to a second arbitrary point on the phase diagram. Thus, there is no reason whatsoever to expect the system to exhibit the same densities. To gain a better understanding on the relationship of density variations on the two regimes, the authors would need to measure densities on multiple combinations of temperatures and salt concentrations. The same can be said for the results on the physical properties of the dense coacervate phases.

R2A4. Thank you for your comments. All measurements of physical properties of coacervates including density, viscosity, interfacial tension were conducted at room temperature (20 – 25 °C). We agree that physical properties would be modulated by temperature. However, as it is shown in the following optical density depending on the temperature, the temperature dependent difference in density between 20 and 25 °C is much smaller than the salt dependent difference in density between low and high salt (Fig. 4c,d, low salt: ~314mg/mL, high salt: ~1908mg/mL), so it can be shown that the physical properties can be greatly altered by hydrophobic interaction. In this study, we'd like to note that arginine contributes to the physical properties of arginine-rich coacervates with its hydrophobicity. Thus, we believe that comparing the physical properties between low and high salt at a fixed temperature is sufficient to state our conclusion.

Temperature-dependent optical density of low and high salt HA-Protamine coacervation.

Fig. 4c and d. Concentrations of hyaluronic acid and protamine within dilute and dense phase of low and high salt HA-Protamine coacervation.

R2Q5. When using molecular simulations again to estimate the radii of gyration and end-to-end distance, I wonder which parameters for ions did the authors used. The parameters for ions in force fields are not necessarily well-fitted to give reasonable ion solubilities at room temperature. This is typically not an issue when performing molecular simulations at physiological salt concentrations, which are the standard. However, at the high salt concentrations used here to mimic the reentrant regime, some ions parameters can yield the formation of unphysical crystals. Thus, estimating the ion solubility of the model is crucial.

R2A5. Here the reviewer brings up a forcefield related issue of salt that we were indeed careful about. We used the NaCl forcefield alongside with the TIP4P/2005 forcefield for water that is introduced by Vega et al in the following paper: J. Chem. Phys. 144, 124504 (2016); <https://doi.org/10.1063/1.4943780> . The NaCl molality with above mentioned forcefield is about ~3.5 M. Although this is far from a realistic salt solubility (~6 M), this forcefield is still a reasonable choice for the purpose of our study. As it is shown in the following MD snapshot, at 3.5 M the ions (blue and cyan spherical particles) do not nucleate crystallization.

The representative MD snapshot of the protamine chain at 3.5 M salt concentration resulted from ~600 ns simulation (water is not shown for the sake of clarity).

Added to Methods (Page 32, Line 707):

We use the parametrized NaCl forcefield by Benavides et al, to prevent unphysical salt crystallization.

Reviewer #3 (Remarks to the Author):

The manuscript reports results from experiments in which the physicochemical properties of complex coacervates formed by an arginine-rich or an lysine-rich peptide with an oppositely charged factor. The authors state that a major insight from this work is the importance of difference in hydrophobicity between arginine and lysine for complex coacervation-based phase separation.

Overall the experiments are well done and partially supported by molecular dynamics simulations. However, the study has two major limitations.

R3Q1. *the conclusion regarding the important role of hydrophobicity is most strongly based on the differences in phase separation behaviour of the arginine- and lysine-rich peptide at very high salt. This however cannot be translated back to physiologically relevant salt conditions where it is likely that a combination of different factors (maybe hydrophobicity, but likely also others as reported already in the literature) are important. Thus, the physiological relevance of the current data is questionable.*

R3A1. We thank the reviewer for discussing the physiological relevance of our results. As you pointed out, the high salt condition used in this study is very high compared to the physiological ionic strength. However, this high salt concentration is mimicking the dehydrated condition that features the hydrophobicity properties of arginine. This dehydrated cellular condition that we model using high salt concentrations can be instead achieved by crowded and multiple charged metabolite-rich cellular conditions. In the previous literature, we have shown that polyethylene glycol (PEG), a molecular crowder, can be used to drive protein liquid-liquid phase separation by increasing the dehydration entropy [1]. Furthermore, it has been reported that the electric field of proteins is weakened by transient quinary interaction with lysates in a manner similar to NaCl [2]. In agreement with this, multivalent anions (citrate and tripolyphosphate) have shown to drive protein phase separation [3]. Given the abundance of multi-charged metabolites and the fact that their levels are altered by cellular conditions, such as stress and cell division, the high salt concentration and the reentrant phase behavior in this study are expected to account for cellular protein phase separation under cellular condition with enhanced hydrophobic interactions.

Reference:

[1] Park, S., Barnes, R., Lin, Y. *et al.* Dehydration entropy drives liquid-liquid phase separation by molecular crowding. *Commun Chem* **3**, 83 (2020).

[2] Zhang, N., An, L., Li, J., Liu, Z. & Yao, L.. Quinary Interactions Weaken the Electric Field Generated by Protein Side-Chain Charges in the Cell-like Environment. *J. Am. Chem. Soc.* **139**, 647 (2017).

[3] Kim, H., Jeon, B.-J., Kim, S., Jho, Y. & Hwang, D. S.. Upper Critical Solution Temperature (UCST) Behavior of Coacervate of Cationic Protamine and Multivalent Anions. *Polymers* **11**, 691 (2019)

Added to Discussion (Page 21, Line 481):

...The use of very high salt concentration in this study well exceeds physiological ionic strength, but is serving as a tool to study hydrophobic properties of arginine. High salt conditions mimic dehydrating conditions frequently found in the crowded cellular environment. Similar dehydration condition as modeled in this study can be achieved by using molecular crowders or by using high concentrations of multi-charged metabolites found under cellular conditions (ref.1-3). Given the abundance of multi-charged metabolites and the fact that their levels are altered by cellular conditions, such as stress and cell division, the high salt concentration and the reentrant phase behavior in this study are expected to account for cellular protein phase separation under cellular condition with enhanced hydrophobic interactions.

R3Q2. *Several studies already have shown the different roles of arginine and lysine for peptide/protein phase separation. This also includes for example reference #25, in which it was shown that the viscoelasticity of arginine-based condensates is greater than that of lysine-based condensates. Thus the novelty/broad interest of the results from the current study is limited. The manuscript is more suited for a journal with a focus on physical chemistry (i.e. J Phys Chem).*

R3A2. Although the arginine and lysine have been expected to play different roles in protein phase separation, the role of arginine hydrophobicity in dictating the salt-mediated reentrant liquid-liquid phase separation of proteins has not been demonstrated with combined experiments and atomistic molecular dynamics simulation. In addition, no single study has shown that a simple model protein, purposefully stripped of complexity, can display both

simple and complex coacervation under different conditions, and can show reentrant LLPS behavior. Also, that such system can display both UCST and LCST behavior within one system. These offer mechanistic understanding for the role of the arginine-rich proteins in protein LLPS at the molecular level.

REVIEWER COMMENTS

Reviewer #1 (Remarks to the Author):

The authors have addressed most of my concerns, however there are a couple of remaining issues. The first concerns the relevance of the comparison of protamine and polylysine (PL). While PL constitutes only lysines protamine has other amino acids in the sequence, so it is not obvious that all differences observed are attributed to the difference between lysine and arginine. The authors argue that they use protamine and not poly-arginine because the latter is not soluble so they cannot use experimental evidence for the similarity of protamine and poly-arginine. They did do MD simulations on poly-arginine and found that its conformations distribution is more similar to poly-arginine than to PL and they just added one sentence in p. 17 about this. However there are significant differences in the contact map where protamine has significantly more contacts than poly-arginine and these may also play a role in the differences observed. The authors should discuss this possibility seriously. In the current version the potential role of the other amino acids is practically ignored. Now that the authors have added the FRAP and EPR results on the protamine at high salt concentration which revealed very low mobility the possibility of the formation of a gel arises. This possibility should be addressed as well.

Reviewer #2 (Remarks to the Author):

The authors have addressed all my comments, and I think the paper is interesting and represents a valuable contribution to the field. Therefore, I would like to recommend publication of this paper in Nat Commun.

From the revision, I particularly enjoyed reading their new results and insights in reply to my second question (R2Q2). The only final comment I have is that while their reply to my third question on the INDUS method (R2Q3) is satisfactory, the revised manuscript does not include the discussion on errors, difficulty on convergence, dependency on the probe, etc, which I think is interesting. I think such information would be valuable for the readers (I would definitely be interested in reading about that if I was reading the paper). Thus, I would encourage the authors to add the information in their reply to R2Q3 to the final version of the manuscript.

Response Letter to Reviewers

We would like to thank the reviewers for their time and supportive comments. We have revised the manuscript in accordance with their comments and concerns, as detailed below. The major revisions made are highlighted in yellow in the revised manuscript. We hope that the revised manuscript is now acceptable for publication in *Nature Communications*.

Reviewer #1 (Remarks to the Author):

The authors have addressed most of my concerns, however there are a couple of remaining issues.

RIQ1: *The first concerns the relevance of the comparison of protamine and polylysine (PL). While PL constitutes only lysines protamine has other amino acids in the sequence, so it is not obvious that all differences observed are attributed to the difference between lysine and arginine. The authors argue that they use protamine and not poly-arginine because the latter is not soluble so they cannot use experimental evidence for the similarity of protamine and poly-arginine. They did do MD simulations on poly-arginine and found that its conformations distribution is more similar to poly-arginine than to PL and they just added one sentence in p. 17 about this. However, there are significant differences in the contact map where protamine has significantly more contacts than poly-arginine and these may also play a role in the differences observed. The authors should discuss this possibility seriously. In the current version the potential role of the other amino acids is practically ignored.*

R1A1: We thank the reviewer for these comments. As the reviewer pointed out, protamine contains other amino acids aside from arginine, and these amino acids may alter the protein's behavior as a function of salt concentration. To address this point, we have performed additional MD simulation in which we simulated a pseudo-protamine chain in which all the "Arg residues are substituted by Lys". We refer to this protein as pseudo-protamine-RtoK chain. In the figure below, we present a comparison of the conformations adopted by poly-lysine, poly-arginine, protamine, and pseudo-protamine-RtoK as a function of salt concentration.

As shown in the contact map in Panel A, in the absence of salt, poly-lysine mainly adopts extended conformational states with rare intramolecular contacts between the nonadjacent residues along the sequence. At high salt concentration, poly-lysine features less extended conformations due to the screening of the repulsive electrostatic interaction by the excess salt. Poly-arginine (Panel B), much like protamine (Panel D), features more skewed population toward collapsed conformations compared to poly-lysine, indicating a partial role of hydrophobic interactions in driving compaction. At high salt concentration, poly-arginine and protamine both exhibit a greater degree of compaction and more frequent intramolecular contacts between the nonadjacent residues (Panels B and D).

We now turn to the conformational behavior of the pseudo-protamine-RtoK to check the importance of the heterotypic nature of the chain (Panel C). As indicated in Panel C, the average d_{ee} and R_g of the pseudo-protamine-RtoK are lower than poly-lysine and higher than protamine in the absence of salt. In addition, as can be seen from the contact maps, pseudo-protamine-RtoK has less frequent contact between the nonadjacent residues compared to protamine in the salt-free condition. This behavior becomes more pronounced at high salt concentrations. At high salt concentration, the pseudo-protamine-RtoK shows much higher d_{ee} and R_g than protamine. More importantly, the number of intra-contacts in the pseudo-protamine-RtoK is similar to that of the poly-Lys chain and substantially lower than that of the protamine. Our observation suggests that there is a small reduction in the collective conformational behavior (d_{ee} and R_g) of the pseudo-protamine-RtoK relative to the poly-Lys due to the heterotypic nature of the chain. However, the enthalpic interactions that are strongly present in the protamine chain are mainly absent in the pseudo-protamine-RtoK (as were so in the poly-Lys). This suggests that the enthalpic interaction in the protamine chain is most likely due to the hydrophobicity of Arg rather than the effect of other amino acids of the protamine.

In addition to the conformational properties, the average dewetting free energy of arginine is identical in the poly-arginine and protamine and is by $\sim 0.6k_B T$ lower than that of lysine in poly-lysine (Supplementary Figure 3d). This indicates that Arg in protamine and poly-arginine is more hydrophobic than Lys in poly-lysine at the residue level. The globally averaged hydration properties of Arg are also attributed to the hydrophobicity of Arg rather than other amino acids in protamine.

Therefore, it can be seen that the differences of HA-Protamine and HA-PL coacervates (especially at high salt concentration) in the experiments are mainly attributed to the different hydrophobicity of lysine and arginine.

Conformational properties including the probability distribution of the radius of gyration (R_g), the end-to-end (d_{ee}) distance (a and b) and intramolecular contact map (c and d) of (A) poly-lysine, (B) poly-arginine, (C) pseudo-protamine-RtoK, and (D) protamine without and with salt.

Supplementary Figure 3. **d** Average dewetting free energies in the polyLys, protamine, and polyArg.

Added to Results (Page 17, Line 393):

...This conformational property of protamine is similar to that of poly-arginine (Supplementary Fig. 7 *Right*). In addition, we check the effect of the heterotypic nature of protamine on its conformational behavior by computationally comparing the collective behavior of a pseudo-protamine-RtoK chain, in which all the Arg residues of protamine is substituted by Lys with the poly-Lysine chain. We observed that the d_{ee} and R_g of the pseudo-protamine-RtoK chain is slightly lower than the poly-Lys chain, however, the enthalpic intra-molecular interactions of the pseudo-protamine-RtoK (evidenced by the contact map) is mostly similar to that of the poly-Lys chain (Supplementary Fig. 7 *Left*).

Supplementary Figure 8 has now been amended as follows to add the pseudo-protamine-RtoK data:

Supplementary Figure 8. Conformational properties of pseudo-protamine-RtoK and poly-arginine. a, e and b, f represent the probability distribution of the radius of gyration (R_g) and the end-to-end (d_{ee}) distance without and with salt, respectively. c, g and d, h show the intramolecular contact map without salt and in presence of salt, respectively.

Added to Results (Page 12, Line 260):

...As illustrated in Supplementary Fig. 3b, the dewetting free energy of Arg in the protamine chain can vary along the sequence, depending on the type of the nearby residues. This nonidentical dewetting free energy behavior indicates the occurrence of context-dependent hydrophobicity of a particular residue in a peptide featuring a nonuniform chemical sequence. However, the Arg dewetting free energy remains unchanged in a homopeptide polyArg chain (Supplementary Fig. 3c) that mainly features expanded conformations with minimal structural

complexity. The average dewetting free energy of Arg in protamine and poly-arginine is identical (Supplementary Fig. 3d). This indicates that while the local hydrophobicity of a residue can be different along a complex peptide, its globally averaged dewetting free energy can still be similar to that of the residue in a homopeptide.

Supplementary Figure 3 has been amended to add panel d:

Supplementary Figure 3. a Dewetting free energies per water molecule that are independently calculated in the hydration volume of Lys2, Lys16 and Lys33. b, the same quantity is characterized in the hydration volume of Arg4, Arg16 and Arg33 of protamine chain. c the same quantity is characterized in the hydration volume of Arg4, Arg16 and Arg33 of poly-Arg chain. d Average dewetting free energies in the polyLys, protamine, and polyArg.

RIQ2: Now that the authors have added the FRAP and EPR results on the protamine at high salt concentration which revealed very low mobility the possibility of the formation of a gel arises. This possibility should be addressed as well.

R1A2: As you pointed out, protamine molecules in the high salt HA-Protamine coacervates showed very low mobility in the FRAP and EPR. However, with FRAP and EPR alone, it is difficult to distinguish whether coacervates have fluid- or gel-like properties. The reviewer's question is an important one. Thus, we performed microrheology and coalescence observation to address this question.

Based on the observation of the storage modulus (G') and loss modulus (G'') of high salt HA-Protamine coacervates (Fig. 5), we determined that the high salt HA-Protamine coacervates exhibit the behavior of viscoelastic fluids. Generally, gel shows G' values higher than the G'' , indicative of elastic behavior. Also, those G' and G'' values tend to show weak frequency dependence due to the physical and nonpermanent nature of network structure of gel. In addition, as shown in Supplementary Fig. 4, we observed that high salt HA-Protamine coacervates are merged to form a single drop to minimize surface tension (we calculated the interfacial tension in Table 1 from this coalescence data). Thus, we consider the high salt HA-Protamine coacervates in this study have viscoelastic liquid-like properties, accompanied by slowed molecule movement induced by dense packing or/and jamming.

Added to Supplementary Figure 4:

Supplementary Figure 4. Representative images of merging events of low and high salt HA-Protamine coacervates in time series. Scale bar = 5 μm .

Added to Results (Page 13, Line 302):

...However, high salt HA-Protamine coacervates are still considered to have liquid-like properties as shown in Supplementary Fig. 4.

The sentence has been corrected (Page 14, Line 314):

... In contrast, high salt HA-Protamine coacervates exhibit viscoelastic fluid behavior with a crossover frequency (G' value equals G'' value) at lower frequencies compared to low salt HA-Protamine coacervates.

Reviewer #2 (Remarks to the Author):

The authors have addressed all my comments, and I think the paper is interesting and represents a valuable contribution to the field. Therefore, I would like to recommend publication of this paper in Nat Commun.

From the revision, I particularly enjoyed reading their new results and insights in reply to my second question (R2Q2).

***R2Q1:** The only final comment I have is that while their reply to my third question on the INDUS method (R2Q3) is satisfactory, the revised manuscript does not include the discussion on errors, difficulty on convergence, dependency on the probe, etc, which I think is interesting. I think such information would be valuable for the readers (I would definitely be interested in reading about that if I was reading the paper). Thus, I would encourage the authors to add the information in their reply to R2Q3 to the final version of the manuscript.*

R2A1: We have carefully modified the manuscript accordingly to include all the points that the reviewer makes in this comment.

Added to the SI as the text (not caption) relevant to Supplementary Figure 3:

As shown in Supplementary Figure 3, panel a, the Lys hydrophobicity in the homopolymer poly-Lys does not change across different sites and locations along the peptide. The calculated free energies feature negligible errors, i.e., less than $0.03kT$ (characterized by bootstrapping). In fact, we observe similar behavior if we investigate the Arg residue hydrophobicity in the homopolymer polyArg (Supplementary Figure 3 panel c). In contrast, we see that the Arg dewetting free energy is a context dependent parameter in the protamine chain that is composed of different types of residues. This indicates that depending on the type of the adjacent residues, the hydrophobicity of Arg can change. We note however that the average dewetting free energies (over the three investigated residues: 4, 16, 32) in the polyArg and protamine peptides are still identical (Supplementary Fig 3 panel d).

Added to Methods (Page 33, Line 733):

The number of water molecules in the hydration shell of each residue can be conformation dependent, and it can also change by the definition of the hydration volume itself. Here, we chose the size of the spherical probe volume of every heavy atom in such a way that we have at least two-three layers of water molecules in the hydration shell (the first coordination shell). We note that a careful selection of the probe volume is important; indeed, if the spherical probe volumes are very large, the resulting hydration volume would be less relevant to the surface of the target residue and the dewetting free energy can be largely affected by the errors due to the hydration water fluctuation.

Added to the SI as the text (not caption) relevant to Supplementary Figure 11:

Considering that the average normalized water fluctuation (to N_0 , the total hydration water at equilibrium; for Arg₁₆ in protamine $N_0 = 44$) as shown in Supplementary Figure 10, is ~ 0.03 across different biased N^* , the effective dewetting free energy fluctuation per-water molecule is only ~ 0.06 kT. It is to be noted that ~ 0.06 kT is 0.1 of the dewetting free energy difference between Arg and Lys.

Additional revision on EPR

The paragraph corresponding to the EPR results has been corrected as follows for clarity (Page 14, Line 324):

...Next, the rotational dynamics of the polymer molecules was evaluated by continuous-wave Electron Paramagnetic Resonance (cw-EPR) lineshape analysis of spin labels tethered to the biopolymer surface. To achieve spin labeling, the amine groups of protamine and ϵ PL were functionalized with the nitroxide radical, here 4-carboxy TEMPO. Consequently, the spin labels are statistically distributed throughout the ϵ PL chain, but are located exclusively at the N-terminal of protamine (Supplementary Fig. 5). The spin-labeled SL-protamine or SL- ϵ PL were mixed with hyaluronic acid (HA), and upon LLPS, centrifuged down to isolate the dilute and dense phase. The rotational correlation time (t_{corr}) was derived from cw-EPR lineshape analysis assuming isotropic rotational motion of the spin label. On SL- ϵ PL in the HA- ϵ PL dense phase t_{corr} was found to be 454 ps, representing dynamic liquid state behavior (Fig. 6b *Top*). Consistent with the FRAP results, the motion of SL-protamine in low salt HA-Protamine coacervate phase was found to be also consistent with liquid state behavior, as represented with $t_{\text{corr}} = 151$ ps (Fig. 6b *Middle*). The t_{corr} value is lower (more dynamic) for SL-protamine despite the higher polyelectrolyte concentration in the HA-Protamine, compared to the HA- ϵ PL, coacervate phase. The absolute value for t_{corr} of SL-protamine may be lower than for SL- ϵ PL because the spin label of SL-protamine is exclusively tethered to the protamine chain ends that are more dynamic than the middle of a biopolymer chain. Overall, SL-protamine and SL- ϵ PL are highly mobile, representing behavior of a liquid coacervate phase formed under low salt conditions. In contrast, the rotational motion of SL-protamine in high salt HA-Protamine coacervates is significantly hindered. This is reflected in the cw-EPR lineshape whose major population (a minor fast motion population is present, likely originating from residual dilute phase) presents a spectral feature in the “rigid limit” for the spin labels of SL-protamine, despite their positions at protamine chain ends (Fig. 6b *Bottom*). This observation is consistent with the dramatically higher viscosity and viscoelastic property of the dense HA-Protamine coacervate phase formed under high salt conditions. These results further validate that enhanced hydrophobic interactions from arginine impart material properties to arginine-rich phases of LLPS (Supplementary Fig. 6).

REVIEWERS' COMMENTS

Reviewer #1 (Remarks to the Author):

The authors have made significant efforts to address my remaining concerns and have no further comments. I recommend accepting the manuscript as it.

Response Letter to Reviewers

Reviewer #1 (Remarks to the Author):

The authors have made significant efforts to address my remaining concerns and have no further comments. I recommend accepting the manuscript as it.

We thank the reviewer for the thoughtful and thorough review.